# The RGD-binding integrins αvβ6 and αvβ8 are receptors for mouse adenovirus-1 and -3 infection

**Manuela Bieri[1,2⊛], Rodinde Hendrickx[1,2⊛], Michael Bauer[1], Bin Yu[3], Tania Jetzer[1], Birgit Dreier[4], Peer R. E. Mittl[4], Jens Sobek[5], Andreas Plückthun[4], Urs F. Greber[1], Silvio Hemmi[1]***

**1** Department of Molecular Life Sciences, University of Zurich, Zurich, Switzerland, **2** Molecular Life Sciences Graduate School, ETH and University Of Zurich, Switzerland, **3** National Engineering Laboratory for AIDS Vaccine, School of Life Sciences, Jilin University, Changchun, China, **4** Department of Biochemistry, University of Zurich, Zurich, Switzerland, **5** Functional Genomics Center Zurich, Eidgenössische Technische Hochschule (ETH) Zurich and University of Zurich, Zurich, Switzerland

⊛ These authors contributed equally to this work.
* silvio.hemmi@mls.uzh.ch

**Data Availability Statement:** All relevant data are within the manuscript and its Supporting Information files, or where indicated, are available from Mendeley Data (https://data.mendeley.com/),

## Abstract

Mammalian adenoviruses (AdVs) comprise more than ~350 types including over 100 human (HAdVs) and just three mouse AdVs (MAdVs). While most HAdVs initiate infection by high affinity/avidity binding of their fiber knob (FK) protein to either coxsackievirus AdV receptor (CAR), CD46 or desmoglein (DSG)-2, MAdV-1 (M1) infection requires arginine-glycine-aspartate (RGD) binding integrins. To identify the receptors mediating MAdV infection we generated five novel reporter viruses for MAdV-1/-2/-3 (M1, M2, M3) transducing permissive murine (m) CMT-93 cells, but not B16 mouse melanoma cells expressing mCAR, human (h) CD46 or hDSG-2. Recombinant M1 or M3 FKs cross-blocked M1 and M3 but not M2 infections. Profiling of murine and human cells expressing RGD-binding integrins suggested that αvβ6 and αvβ8 heterodimers are associated with M1 and M3 infections. Ectopic expression of mβ6 in B16 cells strongly enhanced M1 and M3 binding, infection, and progeny production comparable with mαvβ6-positive CMT-93 cells, whereas mβ8 expressing cells were more permissive to M1 than M3. Anti-integrin antibodies potently blocked M1 and M3 binding and infection of CMT-93 cells and hαvβ8-positive M000216 cells. Soluble integrin αvβ6, and synthetic peptides containing the RGDLXXL sequence derived from FK-M1, FK-M3 and foot and mouth disease virus coat protein strongly interfered with M1/M3 infections, in agreement with high affinity interactions of FK-M1/FK-M3 with αvβ6/αvβ8, determined by surface plasmon resonance measurements. Molecular docking simulations of ternary complexes revealed a bent conformation of RGDLXXL-containing FK-M3 peptides on the subunit interface of αvβ6/β8, where the distal leucine residue dips into a hydrophobic pocket of β6/8, the arginine residue ionically engages αv aspartate215, and the aspartate residue coordinates a divalent cation in αvβ6/β8. Together, the RGDLXXL-bearing FKs are part of an essential mechanism for M1/M3 infection engaging murine and human αvβ6/8 integrins. These integrins are highly conserved in other mammals, and may favour cross-species virus transmission.

doi: 10.17632/vwjnkwxx4z.1. Mendelay data also including excel files that were used to generate graphs, histograms, and calculations.

**Funding:** SH and UFG were funded by an Initial grant from Training Network grant supporting RH "ADVance" from the FP7 of the European Commission (no. 290002ADVance); SH was funded by the Swiss National Science Foundation (31003A_146286) supporting MBi; UFG was funded by the Swiss National Science Foundation 31003A_179256 / 1; MBi was partly supported from Novartis foundation for medical-biological research (16C222). The funders had no role in study design, data collection and analysis, decision to publish, or preparation of the manuscript.

**Competing interests:** The authors have declared that no competing interests exist

## Author summary

HAdV-derived vectors are widely used in gene and oncolytic therapies and vaccination. Increasingly, non-human AdVs with low human seroprevalence are being developed. The characterization of receptors for virions is of key importance for understanding host range, tissue tropism and viral pathogenesis. Murine models are extremely versatile for genetic, immunological, and vector-based studies, but are less established for murine virus infections. Among the three known MAdVs, the infection biology of M1 has been studied to some extent, but M2 and M3 have remained largely unexplored. Here we identified αvβ6/αvβ8 integrins as receptors for M1 and M3 fiber knobs. The results have implications for viral pathogenesis and the development of therapeutic adenoviral vectors in autologous model systems.

## Introduction

Adenoviruses (AdVs) are non-enveloped double-stranded DNA viruses, widespread in the vertebrate kingdom. Members of the genus *Mastadenovirus* include HAdVs and MAdVs [1]. HAdVs typically cause self-limiting disease in the respiratory or gastrointestinal tracts, as well as conjunctiva. They can cause disease with more severe outcomes in immunocompromised patients, possibly involving reactivation of persistent virus [2]. Genome sequencing efforts have expanded the number of HAdVs to more than 100 types, classified into species A to G (http://hadvwg.gmu.edu/). AdVs can be engineered to carry large transgenes, which is an important feature for vaccine vectors, gene therapy and oncolytic applications [3–5].

AdVs infect most cell types upon binding of the protruding fiber knob (FK) to a protein receptor on the cell surface [6–8]. However, there are notable exceptions, such as the binding of hexon of HAdV-D56 to CD46, and the HAdV-C5 (H5) hexon hypervariable loop 1 binding to the murine scavenger receptor A6 [9,10]. For most AdVs the amino acid composition and structure of the FK specifies a high affinity interaction to receptors on the cell surface, including CAR, CD46, DSG-2 and sialic acid. Upon cell binding, HAdVs typically require a secondary receptor for endocytic uptake. This is usually mediated by the arginine-glycine-aspartate (RGD) sequence in an exposed loop of the penton base to active state αvβ3/αvβ5 integrins, followed by outside-in signals, which are critical for stimulating virion endocytosis and endosomolysis [8,11–16]. Endosomal escape also involves the exposure of protein VI upon mechanical cues from motor-driven movements of CAR, and sphingolipid conversion in the plasma membrane, as demonstrated with HAdV-C [14,17,18]. The importance of FK binding to a cell surface receptor for infection has been illustrated by numerous observations, for example with cells lacking fiber receptors or with CAR-blind HAdVs [19]. The use of H5 with a cysteine-constrained RGD-motif displayed in the FK showed that although this virus binds with high efficiency to CAR-negative cells, it infects cells less efficiently than CAR-positive sister cells [17]. Notably, viral drifts on CAR and the presence of integrins on the cell surface are key for the timely exposure of the membrane-lytic protein VI underlying the importance of the dual receptors in H5 entry and infection.

While more than 100 HAdV types have been described, there are only three MAdVs types known, MAdV-1 (M1), M2, and M3 [20]. M1/M2 were among the first nonprimate AdVs identified in the 1960s, isolated from common house mice [21,22], whereas M3 was isolated more recently from the striped field mouse [23]. Phylogenetically, M1 and M3 are more closely related to each other than to M2 [23,24]. MAdVs have characteristics in common with HAdVs

and have been studied in acute and persistent infection using autologous models, with impact on vectorology in human gene therapy and vaccination [20,23,25–30].

M1 causes life-threatening infection in suckling mice and death of adult mice with particular genetic backgrounds [21,31–36]. It infects endothelial cells in the central nervous system (CNS), breaks down the blood-brain-barrier and disseminates to astrocytes and microglia [32,37–39]. In contrast, neither M2 nor M3 infections are phenotypically symptomatic. M2 propagation is restricted to the intestine and renders infected animals immune against a subsequent virus challenge [40,41]. In a side-by-side experimental infection study, M3 showed a strong preference for myocardial tissue and was undetectable in the brain, whereas M1 was rather homogenously distributed in all organs, including the brain [23].

All three MAdVs lack an RGD sequence in their penton base [23,24]. Yet, both M1 and M3 FKs harbor a stretch of 50 amino acid (aa) residues with an RGD-sequence not found in HAdV FKs. This stretch is equivalent to the DE loop of HAdV [42] and is likely situated on the surface of the FK. A range of experimental data including blocking studies with soluble FK proteins, RGD-peptides, anti-αv integrin antibodies, cell treatment with EDTA and integrin knock-out mouse embryonic fibroblasts suggested αv integrins as attachment and entry receptors binding the RGD motif of the FK-M1 [43]. In addition, experiments with soluble heparin, heparinase-treated cells, and fibroblasts deficient in heparan sulfate proteoglycan (HSPG) biosynthesis suggested HSPG as an attachment factor [43,44]. M1 was also shown to associate with coagulation factors IX/X, albeit without affecting virus binding to cells [45].

Integrins are highly conserved heterodimeric transmembrane proteins consisting of α/β-subunits. They mediate bi-directional communication between the extracellular and intracellular space, and are involved in cell migration, proliferation, differentiation, as well as hemostasis, immune response, development, and cancer [46–49]. Under physiological conditions, most integrins adopt a bent-closed or extended-closed headpiece conformation with low affinity for ligands. Intercellular tensile forces (inside-out signaling) together with ligand binding (outside-in signaling) induce conformational changes and stabilize integrins in the extended-open conformation with a 700- to 5,000-fold higher affinity for ligands than the extended-closed or bent-closed conformations [50–52]. The activation state of integrins is positively and negatively influenced by $Ca^{2+}$ and $Mg^{2+}$ binding to three different sites, including the metal ion-dependent adhesion site which binds $Mg^{2+}$ (for review see [53]). Binding of the unphysiological $Mn^{2+}$ is thought to shift the equilibrium towards full activation.

Among the 24 mammalian integrin heterodimers, eight recognize the RGD motif of extracellular matrix (ECM) proteins, that is αvβ1, αvβ3, αvβ5, αvβ6, αvβ8, αIIbβ3, α5β1, α8β1 [47,54]. The RGD motif was first discovered in fibronectin, and later in other ECM proteins [46,55]. It protrudes in a β-turn conformation and is anchored to a well-shaped pocket of the activated integrin. One adhesion point of RGD involves the Asp carboxylate contacting the $Mg^{2+}$-dependent adhesion site in the βI domain of β subunit, and the other one the Arg side chain, which forms a salt bridge to an acidic residue in the β-propeller domain of the α subunit. Residues adjacent to RGD seem to provide specificity for the ligand by making additional contacts [56,57]. The notion that M1 uses RGD-binding integrin(s) for infection, and the wealth of knowledge on receptor use of human and nonhuman AdVs prompted us to investigate the nature of MAdV receptors.

We adopted a stringent definition of a virus receptor, contrasting attachment factors and facilitators [58]. We consider a receptor to be a protein, sugar, or lipid molecule, which makes direct physical contact to a virion component on the cell surface, and thereupon triggers the uptake of the virus particle (or at least the viral genome) into the cell, followed by either infection or inactivation. Receptors stand in contrast to attachment factors, which can bind to the virus particle but alone do not lead to infection or virion clearance. They also stand in contrast

to a third class of cell surface molecules, so called facilitators, which assist the function of receptors for the virion. These distinctions are increasingly important, as the cell-to-cell variability of virus infections involves not just the cell internal state [59], but also the abundance of cell surface molecules and their interactions with virions (for a recent review, see [60]). To fulfill stringent receptor criteria, we used viral and host genetic engineering, biochemical and cell biological assays as well as molecular modeling, and demonstrate that M1/M3 use RGD-motifs in their FKs to bind with high affinity to αvβ6 and αvβ8 integrins for infection of murine and human cells.

## Results

### Generation and characterization of MAdV vectors

We generated a set of nine novel recombinant GFP reporter viruses, five of them from MAdVs, some with E1A deletions (Table 1 and S1 and S2 Figs). M1-IX-2A-GFP (in short M1-IX-G) and M3-IX-2A-GFP (in short M3-IX-G) contain a GFP cassette inserted at the end of the protein IX gene. The 2A sequence causes skipping of one peptide bond during translation, resulting in two distinct proteins, IX and GFP derived from a single mRNA [27,61,62]. Protein IX is expressed from its own intermediate early promoter, leading to low initial transcripts, and increased expression after DNA replication [63]. In the three viruses M1-ΔE1A-G, M2-ΔE1A-G and M3-ΔE1A-G the E1A gene was replaced with the GFP open reading frame (ORF). Of note, all three E1A-deleted mutants were replication-competent in non-complementing mouse cell lines, as originally described for E1A-deleted M1 mutants [64].

The H5-ΔE3B-CG-FK-M1 and H5-ΔE3B-CG-FK-M3 reporter viruses drive GFP expression from the CMV promoter (CG) with a substituted FK of M1 and M3, whereas H5-ΔE3B-CG contains the endogenous FK. All three viruses carry a deletion of the E3B region, replaced by the CG cassette, as originally described for HAdV-C2 (H2) [67]. As expected, these viruses were fully replication-competent in human cells. The fiber chimeric vectors infected human and mouse cells via the M1 and M3 cell surface receptors, respectively. The E1-deleted H5-ΔE1-CG and H3-ΔE1-CG derived from H5 and HAdV-B3 (H3) were used in earlier studies [65,66]. The novel HAdV-B35 (H35)-derived H35-ΔE1-CG virus carries the E4orf6 gene of H5 in place of its own E4orf6 and has enhanced viral growth [68].

**Table 1. Recombinant adenoviruses used in this study.**

| Virus | Short | Replication | FK | IV/ml | Reference |
|---|---|---|---|---|---|
| MAdV-1-IX-2A-GFP | M1-IX-G | Competent | M1 | $2.7 \times 10^{6\ b}$ | This study |
| MAdV-1-ΔE1A-GFP | M1-ΔE1A-G | Competent | M1 | $1.0 \times 10^{7\ b}$ | This study |
| MAdV-2-ΔE1A-GFP | M2-ΔE1A-G | Competent | M2 | $4.4 \times 10^{7\ b}$ | This study |
| MAdV-3-IX-2A-GFP | M3-IX-G | Competent | M3 | $1.0 \times 10^{7\ b}$ | This study |
| MAdV-3-ΔE1A-GFP | M3-ΔE1A-G | Competent | M3 | $0.43 \times 10^{7\ b}$ | This study |
| HAdV-C5-ΔE3B-CMV-GFP-FK-M1 | H5-ΔE3B-CG-FK-M1 | Competent | M1 | $6.8 \times 10^{8\ c}$ | This study |
| HAdV-C5-ΔE3B-CMV-GFP-FK-M3 | H5-ΔE3B-CG-FK-M3 | Competent | M3 | $1.0 \times 10^{8\ c}$ | This study |
| HAdV-C5-ΔE3B-CMV-GFP | H5-ΔE3B-CG | Competent | H5 | $8.0 \times 10^{10\ c}$ | This study |
| HAdV-C5-ΔE1-CMV-GFP | H5-ΔE1-CG | Defective | H5 | $1.1 \times 10^{10\ c}$ | [65] |
| HAdV-B3-ΔE1-CMV-GFP | H3-ΔE1-CG | Defective | H3 | $4.0 \times 10^{9\ c}$ | [66] |
| HAdV-B35-ΔE1-CMV-GFP-E4orf6-HAdV-C5 | H35-ΔE1-CG | Defective | H35 | $1.0 \times 10^{9\ c}$ | This study |

[a] IV, infectious virus

[b] Titer was determined by GFP IFU assay, infecting CMT-93 cells.

[c] Titer was determined by plaque assay, infecting 911 cells.

To identify cell lines yielding maximal output of M1 and M3, we used the M1-ΔE1A-G reporter virus for an initial infection screen of 37 rodent (mainly mouse) and 113 human cell lines, followed by a second screen of 87 mouse cell lines including the M3-ΔE1A-G (S1 Text). Only a few cell lines gave rise to robust GFP signals. The mouse colon cancer line CMT-93 was the most effective one not only for M1/M3 wild type (wt) and most recombinant viruses, but also for M2 wt, in agreement with earlier reports [69]. Other cells proficient for M1 amplification were 3T6, L929 and 3T3 fibroblasts, as described [69–71], although less effective than CMT-93. Among the 37 human melanoma cell lines and primary short-time melanoma cultures [65,72], 11 were susceptible to M1-ΔE1A-G infection, including M000216, SK-Mel-28, DX3, and M980409.

We next measured the kinetics of M1/M2/M3 wt or recombinant M1-IX-G, M1-ΔE1A-G, M2-ΔE1A-G, and M3-IX-G infections of CMT-93 cells by immunoblot analyses. Lysates of M1-infected cells were probed with rabbit anti-hexon protein-M1 antibodies [62], and newly raised rabbit antibodies against E1A-M1, E1B-19K-M1, and intermediate early protein IX-M1, as well as anti-GFP (Fig 1A). The results were validated with rabbit anti-hexon-M2-antibodies [62] and newly raised E1A-M2 antibodies, or rabbit anti-E1A-M3 and FK-M3 antibodies, as well as cross-reactive anti-hexon-M1 antibodies (S3A and S3B Fig). In general, wt and recombinant viruses had similar expression kinetics, although subtle differences were noted. For example, M1-IX-G gave rise to faster expression of all four viral gene products (E1A, E1B-19K, IX, hexon) compared to M1 wt and M1-ΔE1A-G, possibly due to slight differences in the titers of the input viruses. In addition, the processing of the bicistronic fusion protein M1-IX-2A-GFP was not complete, as shown by two processing forms of IX, and additional unexplained forms (denoted by *). Staining with the anti-protein IX-M1 antibodies gave rise to a weak band of processed IX-2A (Mr 14.1 kDa) and a major band of unprocessed IX-2A-GFP (Mr 41 kDa), whereas staining with the anti-GFP antibodies revealed processed GFP (Mr 27 kDa) and unprocessed IX-2A-GFP. Two processing forms were also found in M3-IX-2A-GFP lysates probed with anti-GFP antibodies (S3B Fig). Fading of the actin signal at later infection time points has been noticed before with MAdV-infected cells [62], although mechanisms are unknown.

To further characterize the kinetics and levels of GFP reporter expression, permissive mouse CMT-93 and human M000216 cells were infected with M1-/M3-IX-G, M2-ΔE1A-G, and the fiber chimeric H5-ΔE3B-CG-FK-M1 and -FK-M3, and analyzed by flow cytometry. In CMT-93 cells, all three recombinant MAdV viruses gave rise to strong GFP expression starting at 8 h pi. M2-ΔE1A-G had faster kinetics than M1-IX-G and M3-IX-G (Fig 1B), possibly mimicking E1A, the first protein coding gene expressed in AdV infection [73]. Within 48 h of infection with any of the three viruses, 80–100% of the cells were GFP-positive. In M000216 cells, M3-IX-G-mediated expression reached comparable levels as in CMT-93 cells at 72 h and 24 h pi, respectively, and M1-IX-G levels remained lower (Fig 1C). While M2-ΔE1A-G gave rise to efficient infection of CMT-93, it was much less efficient in M000216 cells. Of note, in both cell types the fiber-chimeric H5-ΔE3-CG-FK-M1 and H5-ΔE3B-CG-FK-M3 gave rise to fast expression kinetics and strong GFP signals, about 5- to 10-fold higher in CMT-93 cells and about 10- to 100-fold higher in M000216 cells, when compared to the M1-/M3-IX-G viruses. This is likely due to the strong CMV promoter and the replication competence of these viruses in human cells. In mouse CMT-93 cells, all five viruses gave rise to cytopathic effects (CPE), eventually reducing the GFP signals. In contrast, the M1 and M3 fiber chimeric human viruses H5-ΔE3B-CG-FK-M1/-FK-M3 gave rise to CPE in human M000216 cells, but not M1 and M3. This is compatible with the notion that M1 fails to replicate in human cells [70], and indicates that FK-M1 and FK-M3 mediate virus entry. Hence, the mouse fiber

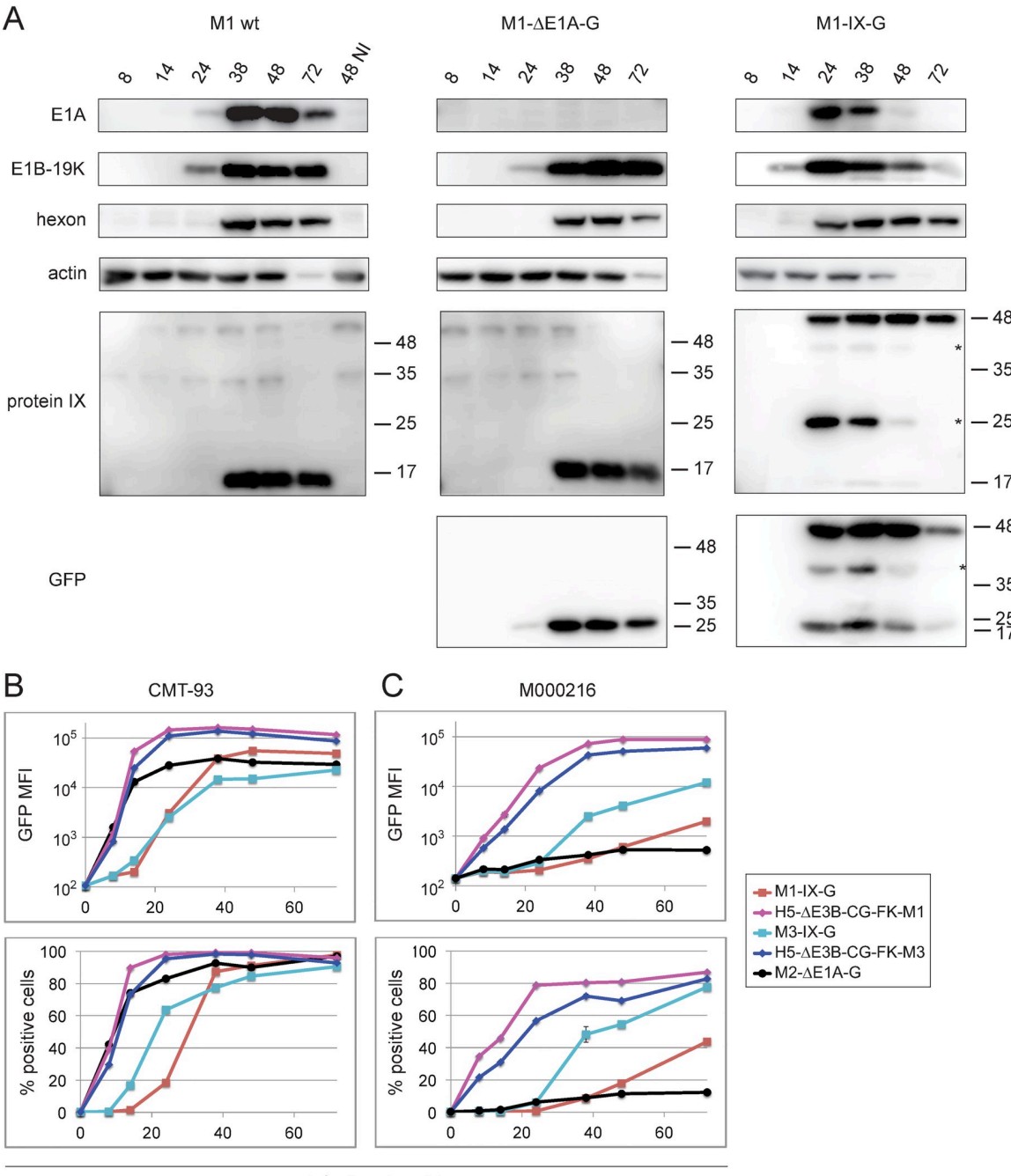

**Fig 1. Infectivity analysis of mouse and human fiber chimeric adenoviruses.** (A) Immunoblot analysis of mouse CMT-93 cells infected with M1 wt, and recombinant M1-ΔE1A-G and M1-IX-G viruses using an MOI of 3. Cell lysate samples were harvested at six time points and analyzed with the indicated rabbit antibodies raised against early E1A-M1, E1B-19K-M1, intermediate protein IX-M1, and the late hexon protein-M1, plus mouse antibodies against GFP and control actin. Staining with protein IX-specific antibodies revealed a weak band corresponding to processed IX-2A (Mr 14.1 kDa), and a major form corresponding to unprocessed IX-2A-GFP (Mr 41 kDa). Staining with GFP-specific antibodies revealed two major processing forms, corresponding to processed GFP (Mr 27 kDa), and the unprocessed IX-2A-GFP, respectively. Both of these stainings gave rise to additional individual protein forms (denoted by *). (B) Mouse CMT-93 cells and (C) human M000216 cells were infected with recombinant M1-/M3-IX-G, M2-ΔE1A-G and fiber-chimeric H5-ΔE3B-CG-FK-M1 and–FK-M3 viruses at an MOI of 3. Cells were harvested at the indicated six time points and GFP intensity (upper panel) and percent infected cells (lower panel) were determined by flow cytometry. Cellular autofluorescence of uninfected cells was included as 0 h infection time point. Data represent triplicates, shown as mean ± SEM.

chimeric viruses H5-ΔE3B-CG-FK-M1 and H5-ΔE3B-CG-FK-M3 represent new tools to explore the tropism of FK-M1-/FK-M3 carrying viruses.

## FK-M1 and FK-M3 cross-block M1 and M3 but not M2 infection

Towards identifying MAdV receptors, we performed blocking studies with the reporter viruses M1-/M3-IX-G and M2-ΔE1A-G as well as H5-ΔE3B-CG-FK-M1/-FK-M3, in combination with recombinant soluble FK-M1, FK-M2, and FK-M3. CMT-93 and M000216 cells were pre-incubated on ice with FKs at 0.26 to 800 ng/ml, followed by infection at 37˚C for 48 h at multiplicity of infection (MOI) 1 and 3 for CMT-93 and M000216 cells, respectively. Both FK-M1 and FK-M3 gave rise to dose-dependent reduction of infection with M1-/M3-IX-G in CMT-93 cells (Fig 2A and 2B). The FK-M1 and FK-M3 concentrations yielding 50% inhibition ($IC_{50}$) of M1-IX-G and M3-IX-G infections were in the range of 2.1 to 4.8 ng/ml, similar to the $IC_{50}$ of ~22 ng/ml for FK-H5 inhibition of H5-ΔE1-CG in human A549 cells [74], but not MAdVs with up to 800 ng/ml (Figs 2A–2E and S4A–S4E and Table 2). FK-M1 and FK-M3 blocked H5-ΔE3B-FK-M1/-FK-M3 at $IC_{50}$ of 3.5 to 5.6 ng/ml (Figs 2C and S4B and Table 2). They blocked M1-IX-G and M3-IX-G infections (MOI 3) of human M000216 cells at $IC_{50}$ values roughly 30-fold higher than for CMT-93 cells without effects from the FK-H5 control protein (Figs 2E and S4E and Table 2).

FK-M2 had no effect on M1-IX-G and M3-IX-G infections (Fig 2A and 2B). Intriguingly, this batch of FK-M2 also failed to block M2-ΔE1A-G infection (Fig 2D), although the DNA construct encoding this protein was identical to one which was recently shown to block M2 infection [75]. We tested two additional FK-M2 constructs with varying lengths of the flanking fiber shaft sequence (see Materials and Methods), but none of them blocked M2-ΔE1A-G (data are deposited at Mendeley Data). Of note, exclusion chromatography coupled to multi-angle light scattering (SEC-MALS) of the purified FK-M2 revealed a discrepancy in the molar mass of the trimeric FK-M2, which was ~14% lower than expected (see Materials and Methods, S5 Fig and S1 Table). This discrepancy might explain the lack of inhibition. The highest concentrations of the FK-M1/-M3 weakly inhibited M2-ΔE1A-G infection, which currently remains unexplained.

Regardless, to further explore the specificity of the FK-M1 and FK-M3 inhibitions, we incubated M1 and M3 viruses with two newly generated rabbit anti-FK-M3 and anti-FK-M2 sera in serial dilutions ranging from 1:1,250 to 1:781,250 and added the mixture to CMT-93 cells at 37˚C for 48 h at MOI 1. The rabbit anti-FK-M3 antiserum gave $IC_{50}$ neutralization at ~1:575,000 and 1:275,000 dilutions for M3-IX-G and H5-ΔE3B-FK-M3, respectively (Fig 2F). Neither the rabbit anti-FK-M2, nor a control rabbit anti-H3-FK serum neutralized M3-IX-G. The rabbit anti-FK-M3 antiserum inhibited M1-IX-G at $IC_{50}$ of 1:4,045 (Fig 2G), suggesting a moderate serological cross-reactivity between the fibers of M1 and M3. The rabbit anti-FK-M2 antiserum showed an $IC_{50}$ value of ~1:6,802 dilution for M2-ΔE1A-G infection, yet we obtained no appreciable neutralization of M2-ΔE1A-G with the rabbit anti-FK-M3 or anti-FK-H3 sera (S6 Fig). Together, the data show that FK-M1 and FK-M3 cross-compete with M1 and M3 infections, and the cross-neutralization of M1 by the anti-FK-M3 antiserum is consistent with the previously suggested close genetic relationship of M1 and M3 viruses [24]. Our results imply that M1 and M3 use identical or similar receptors, distinct from the M2 receptor.

## Neither mouse CAR, nor human CD46 or DSG-2 serve as receptors for M1, M2 or M3

Mouse CAR (mCAR) is expressed in multiple tissues and serves as a receptor for the human CAR-binding AdVs in mice [76,77]. It does not, however, bind M1, as shown in mCAR

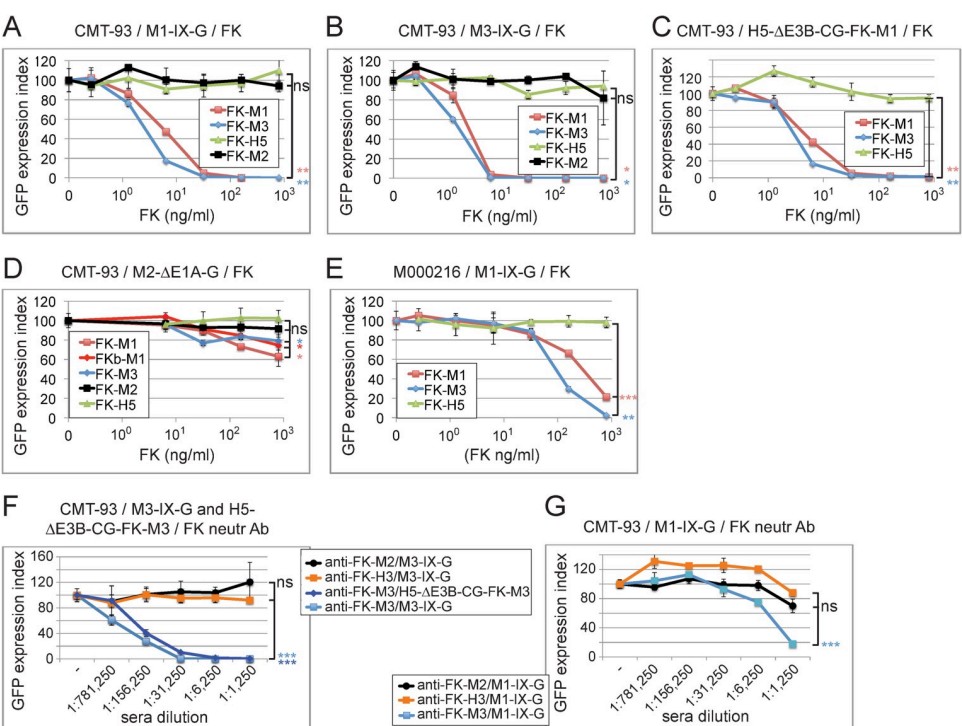

**Fig 2. Inhibition of MAdV and fiber-chimeric GFP reporter virus transduction in CMT-93 and M000216 cells by recombinant MAdV FKs and anti-FK antisera.** (A-D) CMT-93 cells were incubated for 1 h on ice using 5-fold dilution series of the indicated FK proteins starting with 0.8 μg/ml as highest concentration, followed by addition of the different GFP-expressing viruses and transfer to 37˚C for 48 h. The virus input amounted to an MOI of 1 and included M1-IX-G (A), M3-IX-G (B), H5-ΔE3B-CG-FK-M1 (C), M2-ΔE1A-G (D). GFP analysis was performed 48 h pi, and expression index was normalized to FK-H3 control protein. (E) M000216 cells were preincubated and processed as described for CMT-93 cells, except that M1-IX-G was used at an MOI of 3. (F) The M3-IX-G and H5-ΔE3B-CG-FK-M3 viruses were pre-incubated for 1 h at RT with serial 5-fold dilutions of the different antisera ranging from 1:1,250 to 1:781,250, followed by addition of the mixes to CMT-93 cells for 48 h at 37˚C. The rabbit antisera tested were raised against recombinant FK-M3, FK-M2 and FK-H3, respectively. The virus input amounted to an MOI of 1, and samples were processed for analysis as described above. (G) To check for cross-neutralization of the rabbit anti-FK-M3 for M1, M1-IX-G was preincubated with serial dilutions of rabbit anti-FK-M3, -FK-M2 and -FK-H3 sera and further processed as described above. For all experiments, data represent triplicates, shown as mean ± SEM. For highest concentrations of FKs and anti-FK sera, asterisks indicate level of significance for comparison of indicated values (*, $P<0.05$; **, $P<0.005$; ***, $P<0.0005$); ns: not significant ($P>0.05$).

expressing CHO cells [76]. With our set of recombinant MAdVs we tested whether mCAR is a receptor for M2 or M3. Mouse B16 melanoma cells stably expressing FLAG-tagged mCAR showed heterogenous anti-FLAG tag staining, akin to parental B16 cells (Fig 3A), although ~50% of the B16-mCAR cells were strongly positive. Infection of these bulk B16-mCAR cells with MAdVs used at MOI 15 (M1-IX-G, M2-ΔE1A-G and M3-IX-G) failed to give rise to higher GFP expression in B16-mCAR cells compared to B16 parental cells, although all of them efficiently infected CMT-93 cells (MOI 3), and H5-ΔE1-CG infected the B16-mCAR cells as well as CMT-93 cells [78] showing that the FLAG-tagged mCAR was functional (Fig 3B). Weak infection of B16 with H5 was in agreement with the literature [79]. We noticed that M2-ΔE1A-G infection gave a weak GFP expression signal in parental B16, which was not increased in B16-mCAR cells [78,79].

Some viruses may bind to receptors across animal species. We thus tested CHO cell lines stably expressing human CD46 and DSG-2 at levels similar to human A549 cells [74,80] (S7A Fig). None of the three MAdV reporter viruses increased GFP expression in CHO-hCD46 or

**Table 2. IC$_{50}$ values[a] of integrin Ab-, FK-, sITG- and peptide-mediated blocking of MAdV and chimeric HAdV reporter infection.**

| Cell type / mean fluorescence intensity (MFI) of staining with indicated control and integrin antibodies | MOI | Blocking Agent [b] | M1-IX-G | H5-ΔE3B-CG-FK-M1 | M3-IX-G | H5-ΔE3B-CG-FK-M3 | M2-ΔE1A-G | H5-ΔE3B-CG |
|---|---|---|---|---|---|---|---|---|
| CMT93 mouse colon cancer / con Ab: 304 anti-β6: 4,599 anti-β8: 357 | 1 | Ab-β6 | 124 | 138 | 95 | 119 | >800 | >800 |
| | 1 | Ab β8 | >800 | >800 | >800 | >800 | >800 | |
| | 1 | FK-M1 | 4.8 | 5.6 | 3.5 | 5.4 | > 800 | |
| | 1 | FK-M3 | 3.6 | 4.1 | 2.1 | 3.5 | > 800 | > 800 |
| | 1 | FK-H5 | >800 | >800 | >800 | >800 | >800 | 22 |
| | 1 | sITG mαvβ3 | >800 | | >800 | | | |
| | 1 | sITG mαvβ6 | 80 | | 59 | | | |
| | 1 | sITG mαvβ8 | >800 | | >800 | | | |
| | 1 | sITG hαvβ6 | 83 | | 52 | | | |
| | 1 | sITG hαvβ8 | >800 | | >800 | | | |
| | 1 | A20FMDV2 | 6.3 | 4.7 | 1.1 | | >5,000 | |
| | 1 | A20FMDV2-E | >5,000 | >5,000 | 3,770 | | > 5000 | |
| | 1 | A20M1 | 26 | 30 | 5.3 | | > 5000 | |
| | 1 | A20M3 | 7.4 | | 1.3 | | | |
| B16-mβ6 mouse melanoma / con Ab: 142 anti-β6: 3,992 anti-β8: 150 | 3 | Ab-β6 | 150 | 551 | 124 | 201 | | |
| | 3 | Ab-β8 | >800 | >800 | >800 | >800 | >800 | |
| B16-m β8 / con Ab: 123 anti-β6: 130 anti-β8: 3,561 | 3 | Ab-β6 | >800 | >800 | >800 | >800 | | |
| | 3 | Ab-β8 | 61 | 95 | 17 | 75 | | |
| M000216 human melanoma / con Ab: 219 anti-β6: 259 anti-β8: 9,052 | 3 | Ab β6 | >800 | >800 | >800 | >800 | | >800 |
| | 3 | Ab β8 | 73 | 120 | 13 | 64 | | >800 |
| | 3 | FK-M1 | 393 | | 112 | | | |
| | 3 | FK-M3 | 116 | | 68 | | | |
| | 3 | FK-H5 | >800 | | >800 | | | |
| | 3 | sITG mαvβ3 | >800 | | >800 | | | |
| | 3 | sITG mαvβ6 | 11 | | 4.5 | | | |
| | 3 | sITG mαvβ8 | >800 | | >800 | | | |
| | 3 | sITG hαvβ6 | 7.6 | | 2.3 | | | |
| | 3 | sITG hαvβ8 | >800 | | >800 | | | |
| | 3 | A20FMDV2 | 898 | | 165 | | | |
| | 3 | A20M1 | ~5,000 | | 150 | | | |
| | 3 | A20M3 | 422 | | 17.9 | | | |
| A549-hβ6 human lung carcinoma con Ab: 219 anti-β6: 1,349 anti-β8: 220 | 3 | A20FMDV2 | 110 | | | | | |
| | 3 | A20M1 | 336 | | | | | |
| | | | | | | | | |

[a] Data correspond to blocking experiments shown in Figs 2 and 6 and S4, S8, S9 and S11, respectively.

[b] IC$_{50}$ concentrations of antibodies, FKs and sITGs are in ng/ml; concentration of peptides are in nM

CHO-hDSG-2 compared to CHO parental cells (MOI 50), unlike H3-ΔE1-CG and H35-ΔE1-CG infections of CHO-hCD46, CHO-hDSG-2 and A549 cells (MOI 10) (S7B Fig). Collectively, these results support the notion that M1 does not bind to mCAR [76] and extend this conclusion to M2 and M3, making it unlikely that M1/M2/M3 are making cross-species use of the major HAdV receptors CAR, CD46, and DSG-2.

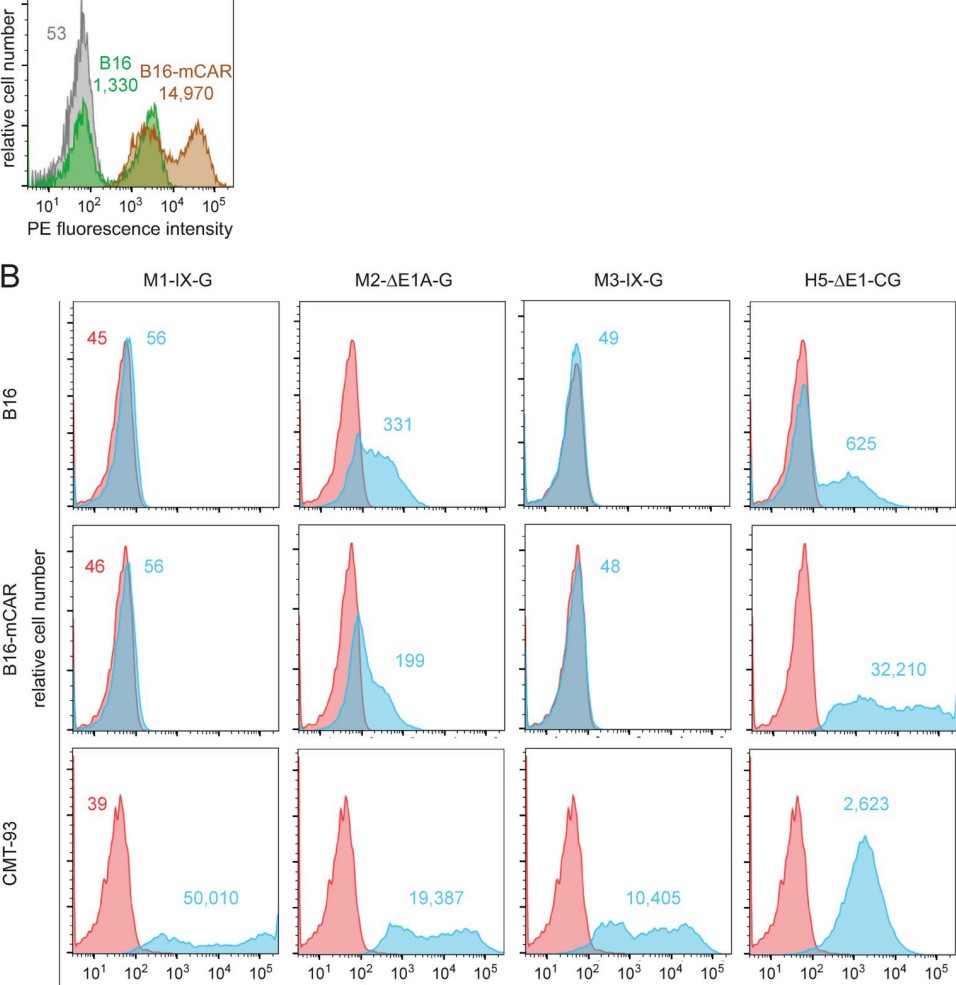

**Fig 3. Evaluation of mCAR as receptor for M1/M2/M3.** (A) The green and brown histograms show cytofluorometric analysis of FLAG-tag expression levels in parental B16 and B16-mCAR cells, consisting of B16 cells stably expressing N-terminal FLAG-tagged mCAR, respectively. The grey histogram shows background staining of B16-mCAR cells using a matched isotype control. Numbers indicate MFI values of specific or control antibodies. (B) Parental B16 cells with known low, and CMT-93 cells with high sensitivity for H5-ΔE1-CG, plus B16-mCAR cells were infected with M1-IX-G, M2-ΔE1A-G, M3-IX-G and H5-ΔE1-CG using an MOI of 15 for both B16 cell types, and an MOI of 3 for CMT-93 cells. Cells were analyzed by flow cytometry 48 h pi. MFI values for GFP in red and blue histograms are from uninfected and infected cells, respectively.

## Profiling of integrin expression in murine and human cells identifies αvβ6/αvβ8 candidate receptors for M1 and M3 infections

To evaluate which RGD binding integrins could serve as receptors for M1 and M3 infections, we compared their expression levels in cells susceptible or resistant to M1 or M3. Flow cytometry analysis of M1/M3-sensitive mouse and human cells with 11 human and mouse integrin-specific antibodies included the highly sensitive CMT-93, intermediate sensitive L929 and 3T6, and resistant B16 cells (Fig 4A), as well as sensitive M000216, SK-Mel-28, and low-sensitive A549 and 911 human cells (Fig 4B). In CMT-93 and L929 cells αvβ6 relative MFI levels

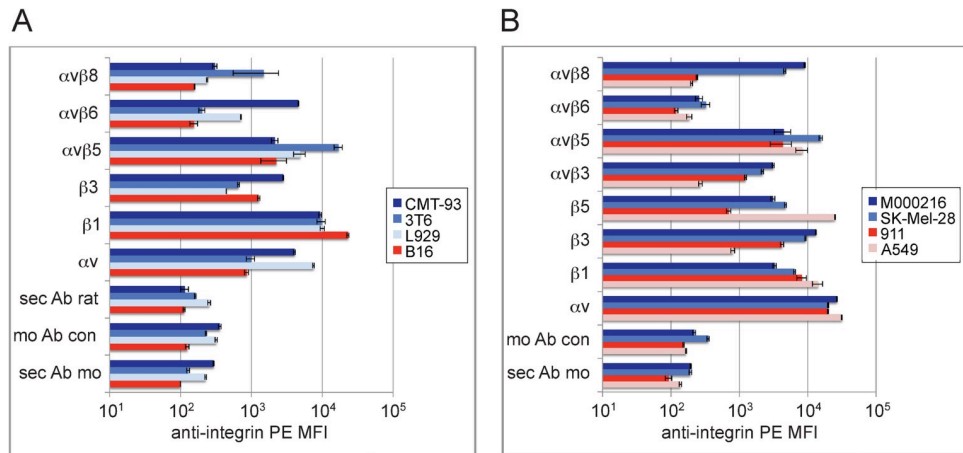

**Fig 4. Profiling analysis of integrin expression levels in mouse and human cells known for sensitivity or resistance to M1/M3.** (A) Monoclonal antibodies specific for the indicated integrins were used to stain mouse cells including highly M1-/M3-sensitive CMT-93, moderately sensitive 3T6 and L929 cells, and resistant B16 cells. (B) Human cells including M1-/M3-sensitive M000216 and SK-Mel-28 cells and resistant A549 and 911 cells were processed as described above. Staining with primary and secondary antibodies was followed by flow cytometry analysis. Control stainings included incubation with goat anti-rat secondary antibodies only (sec Ab rat), mouse IgG1 isotype control plus goat anti-mouse secondary antibodies (mo Ab con), and goat anti-mouse secondary antibodies only (sec Ab mo). Data represent triplicates, shown as mean ± SEM.

amounted to 4,599 and 710, respectively (Fig 4A and Table 2). In 3T6, the αvβ8 MFI levels amounted to 1,483. With human cells, sensitivity correlated with αvβ8 and αvβ3, but again not with αvβ5 or β1 integrins. In M000216 and SK-Mel-28 cells the αvβ8 levels were 9,052 and 4,447, respectively (Fig 4B and Table 2). This suggested that for mouse cells, integrin receptor candidates comprised αvβ6 and αvβ8, but not αvβ5, β1- or β3-containing integrin heterodimers. The β6 and β8 integrins are functionally and genetically more related to each other than to other β subunits [81–84], and were thus further investigated as receptor candidates for M1 and M3.

## β6 and β8 integrin expressions in mB16 cells enhance M1/M3 binding, infection, and progeny production

We prepared mouse B16 cells and human A549 cells stably expressing murine and human β6/β8 integrin subunits. These β-chains exclusively pair with endogenous αv and are co-transported to the cell surface. The parental cell lines had previously been shown to be insensitive to M1/M3 infection (Fig 4A and 4B). They express background levels of αvβ6 and αvβ8 integrins. Analyses of cell surface αvβ6 gave MFI levels of 3,992 for B16-mβ6, comparable to CMT-93 cells, while αvβ8 levels in B16-mβ8 cells were 3,561, about 2.4-fold higher than in mouse 3T6 cells, comparable to human SK-Mel-28 and M000216 cells (Figs 4A, 4B and 5A and Tables 2 and 3). Of note, both anti-β6 and anti-β8 antibodies are species cross-reactive and were used under identical settings making the results comparable between human and mouse cell lines. Stable hβ8 cells could not be obtained, possibly because the expression of β8 inhibits cell growth both in vitro and in vivo, as reported earlier [85]. Yet, we obtained stable expression of hβ6 in A549 cells with MFI of 1,349 (αvβ6), only 3.5-fold lower than in mouse CMT-93 cells (Figs 4A and 5A and Table 2).

 The stable integrin-engineered cell lines were first tested for virus binding using our potent neutralizing rabbit anti-FK-M3 antiserum. Suspended cells were cold-incubated with virus inocula which previously tested positive for infection, followed by staining with anti-FK-M3,

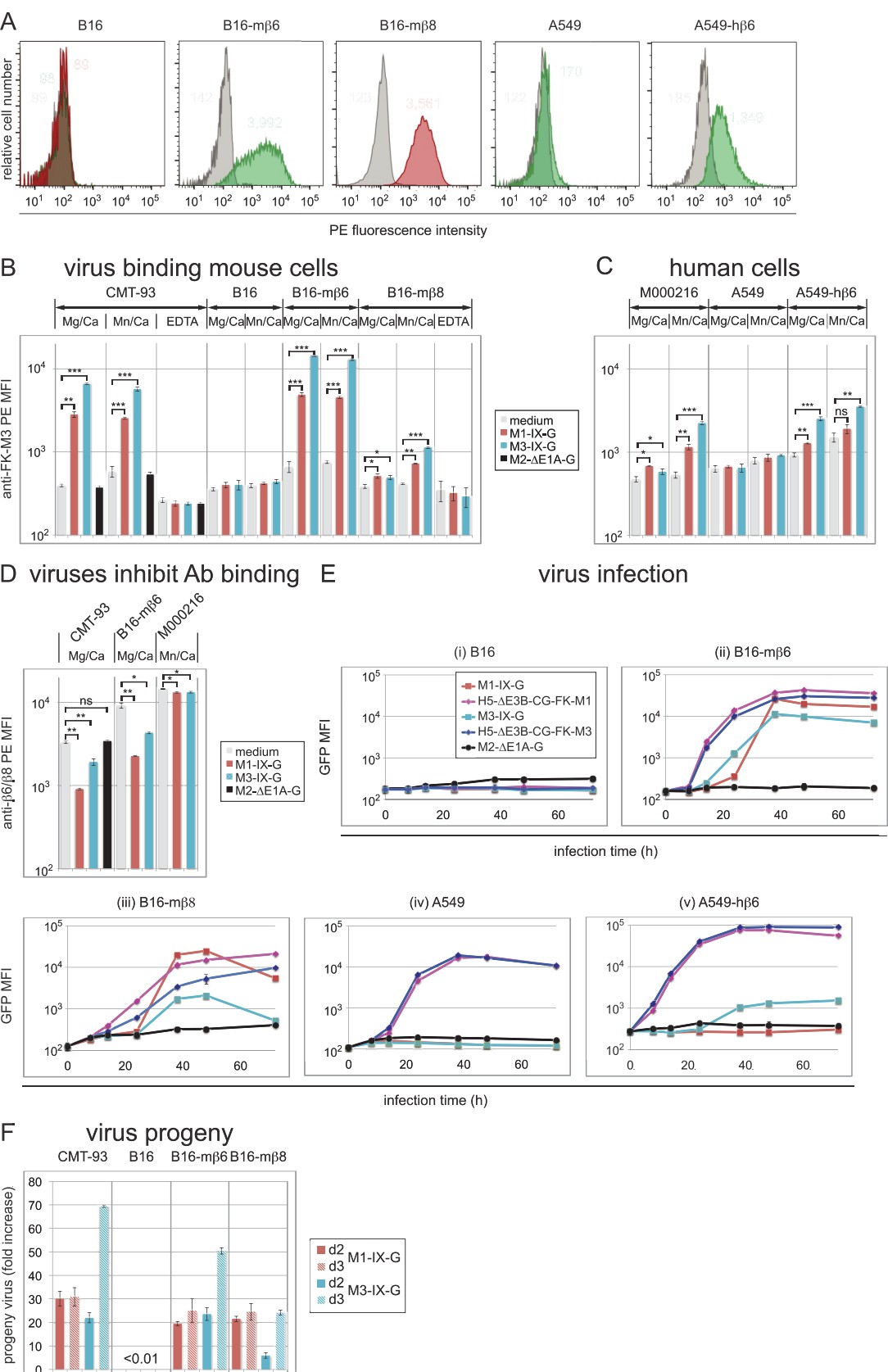

**Fig 5. Virus binding, infection and progeny production in mouse B16-mβ6/8, and human A549-hβ6 cells.** (A) Flow cytometry profiles of B16, B16-mβ6, B16-mβ8, A549 and A549-hβ6 cells. Green and red histograms show β6 and β8 specific staining, respectively, and grey histograms show background staining using an isotype control. Numbers indicate MFI values of specific or control antibodies. (B, C) Virus binding and dependency on divalent ions. Detached mouse (B) and human cells (C) were incubated with control medium (not containing virus) or medium containing the indicated viruses for 1 h on ice, followed by washing and staining with primary rabbit anti-FK-M3 antibodies and secondary fluorescently labeled antibodies for flow cytometry analysis. Incubation/washing buffers were adjusted to contain either $Mg^{2+}/Ca^{2+}$, 1 mM each, 1/0.2 mM $Mn^{2+}/Ca^{2+}$, or EDTA 2.5 mM. (D) Virus-receptor antibody competition experiment. Detached cells were first incubated on ice with control medium or the indicated viruses, followed by washing and incubation with either the anti-αvβ6 antibody (CMT-93 and B16-mβ6 cells), or the αvβ8 antibody (M000216 cells). Subsequently the cells were stained with secondary fluorescently labeled antibodies for flow cytometry analysis. (E) The indicated mouse and human cells were infected with recombinant M1-/M3-IX-G, M2-ΔE1A-G and fiber chimeric H5-ΔE3B-CG-FK-M1/-FK-M3 viruses at an MOI of 3. Cells were harvested at the indicated six time points and GFP intensity (MFI) was determined by flow cytometry. Cellular autofluorescence of uninfected cells was included as 0 h infection time point. (F) For analysis of virus progeny, CMT-93, B16, B16-mβ6 and B16-mβ8 cells were infected with M1-/M3-IX-G using an MOI of 1.5. After 14 h, the cells were thoroughly washed, trypsinized and re-seeded. Virus-containing supernatant samples were collected 48 (d2) and 72 h (d3) pi and used for titration analyses. Based on the virus input, fold increases of progeny virus were calculated. For B16 cells no measurable levels of viruses were detected, which translated to a virus progeny production of less than a factor of 0.01, based on the sensitivity level of this assay. Data in (B) to (F) represent triplicates, shown as mean ± SEM. Asterisks indicate level of significance for comparison of indicated values (*, $P<0.05$; **, $P<0.005$; ***, $P<0.0005$).

and fluorescent secondary antibodies. M1-IX-G and M3-IX-G binding to the αvβ6-positive CMT-93 cells in $Mg^{2+}/Ca^{2+}$ buffer was increased by 7.1- and 16.7-fold, respectively, compared to conditions lacking virus, or by using M2-ΔE1A-G (Fig 5B). Cell-bound M1-IX-G was less prevalent than M3-IX-G, possibly due to lower affinity of M1 to CMT-93 cells than M3, or due to moderate cross-neutralizing activity of the anti-FK-M3 serum against M1. EDTA completely abolished virus binding, in line with the divalent ion-dependent nature of the integrin-ligand interaction. Replacement of $Mg^{2+}$ by $Mn^{2+}$ did not enhance virus binding. With B16-mβ6 cells, a 7.4- and 21.8-fold increase of M1-IX-G and M3-IX-G binding was found in $Mg^{2+}/Ca^{2+}$ buffer, respectively, when compared to control cells in medium, independent of $Mn^{2+}$, and parental B16 cells did not bind M1- or M3-IX-G virions (Fig 5B). The binding of M1- or M3-IX-G to B16-mβ8 cells was weakly but significantly increased by 1.3-fold in $Mg^{2+}/Ca^{2+}$ buffer, and binding of M3-IX-G by 2.7-fold in $Mn^{2+}$ buffer. M1-IX-G and M3-IX-G moderately bound to avβ8-positive M000216 and A549-hβ6 cells, but not to parental A549 (Fig 5C). The modest 2.7-fold binding increase to A549-hβ6 cells in $Mg^{2+}/Ca^{2+}$ buffer was likely due to low β6 expression in these cells and was not enhanced by $Mn^{2+}$. In M000216 cells, the presence of $Mn^{2+}$ increased the binding enhancement by 1.5- and 3.4-fold for M1- and M3-IX-G, respectively. Furthermore, M1/M3 binding to cells reduced the binding of anti-β6- and anti-β8-integrin antibodies to cells, thus providing further evidence that M1/M3 bind to β6 and β8 integrins. M1-/M3-IX-G led to a 3.7- and 1.7-fold reduction in β6-integrin antibody signal on CMT-93 cells compared to control medium without virus or medium with M2-ΔE1A-G (Fig 5D). On B16-mβ6 cells, anti-β6-antibody binding was reduced 4.1- and 2.1- fold by M1- and M3-IX-G, respectively, while with αvβ8-positive M000216 cells a weak but significant 10% reduction of the β8-integrin signal was obtained in $Mn^{2+}/Ca^{2+}$ buffer.

The virion-cell binding results were confirmed with infection studies of CMT-93 and M000216 cells. M1-/M3-IX-G as well as fiber-chimeric H5-ΔE3B-CG-FK-M1/-FK-M3 gave rise to low GFP expression in B16 cells (Fig 5E-i) but strongly increased in B16-mβ6 (Fig 5E-ii), unlike M2-ΔE1A-G. While the M1-/M3-IX-G GFP expression levels were comparable in B16-mβ6 and CMT-93 cells, H5-ΔE3B-CG-FK-M1/-FK-M3 levels were 3.7- and 12.6-fold lower in B16-mβ6 than in CMT-93 cells, possibly reflecting differences in the GFP promoter activities (Figs 1B and 5E-ii). In B16-mβ8 cells, the M1-IX-G virus yielded comparable maximal GFP expression levels to CMT-93 and B16-mβ6 cells, while M3-IX-G gave about 10-fold lower levels, akin to the fiber-chimeric viruses H5-ΔE3B-CG-FK-M1/-FK-M3 viruses (Figs 1B

and 5E-iii). Intriguingly, H5-ΔE3B-CG-FK-M1/-FK-M3 gave rise to strong GFP expression in parental A549 cells (Figs 5E-iv). This was boosted about 5-fold in A549-hβ6 cells, where M3-IX-G but not M1-IX-G increased GFP expression 13-fold (Fig 5E-v). The data strongly suggest that the FK of M1 and M3 use mβ8 and mβ6 for infection, where M3 may make more efficient use of hβ6 than M1.

Finally, we tested if expression of mβ6 and mβ8 in B16 cells increased virus progeny production. M1-IX-G progeny increased 20-30-fold over input (48 or 72 h pi) in either CMT-93, B16-mβ6 or B16-mβ8 cells (Fig 5F). M3-IX-G progeny in CMT-93 and B16-mβ6 reached similar levels as M1-IX-G. In B16-mβ8 cells, however, M3-IX-G progeny production was about 3-fold lower compared to CMT-93 and B16-mβ6 cells, while parental B16 cells did not contain measurable levels of infectious particles.

In summary, the expression of mβ6 in mouse B16 cells robustly enhanced binding, infection and progeny production of M1 and M3 reporter viruses, while B16-mβ8 cells bound M1 or M3 viruses less effectively and gave lower M3 reporter expression and progeny formation. In contrast, M1-IX-G transduction and progeny production in B16-mβ8 cells were not different from B16-mβ6 cells, suggesting that the somewhat reduced virus binding could be compensated for downstream of entry. M1 and M3 appear to make distinct use of αvβ6 and αvβ8 integrins.

## Anti-αvβ6/8 integrin antibodies inhibit M1 and M3 binding and infections

We first tested whether the function-blocking monoclonal anti-β6 antibody [86] and anti-β8 antibody [87] blocked M1/M3 binding and infection of mouse and human cells. Pre-incubation of mCMT-93 cells with anti-β6 antibody resulted in 3.7- and 9.4-fold reduction of M1 and M3 binding, respectively, while pre-incubation of hM000216 cells with the anti-β8 antibody reduced binding 1.8- and 1.4-fold, which was further ~2-fold reduced in the presence of $Mn^{2+}$ (Fig 6A). Antibody titration experiments in the β6-expressing CMT-93 or B16-mβ6 cells showed that the anti-β6 antibody (but not the anti-β8 antibody) strongly reduced infection by M1- and M3-IX-G and H5-ΔE3B-CG-FK-M1 and -FK-M3 viruses (MOI 1 or 3), yielding $IC_{50}$ values in the range of 95 to 551 ng/ml (Figs 6B, S8A and S8B and Table 2). In contrast, no inhibition of M2-ΔE1A-G and H5-ΔE3B-CG infections was observed (Fig 6B). With the β8-expressing M000216 or B16-mβ8 cells, the anti-β8 antibody inhibitions gave $IC_{50}$ values of 13 to 120 ng/ml (Figs 6C and S8C and Table 2), whereas the anti-β6 antibody had no appreciable effect in M000216 cells (S8D Fig). We noticed that with the β8-expressing cells there was a trend towards higher potency of the anti-β8 antibody in blocking the M3-IX-G infection compared to the M1-IX-G infection, an effect that was not seen with the fiber chimeric H5-ΔE3B-CG-FK-M1/-FK-M3 viruses, however, and thus remains unexplained (Figs 6C and S8C and Table 2).

Single antibody and FK concentrations (0.8 μg/ml of each) were found to reduce M1- and M3-IX-G and H5-ΔE3B-CG-FK-M1 and -FK-M3 infections (MOI 3 or 9) of αvβ6-positive mL929 and m3T6 cells, but the reduction was less pronounced compared to CMT-93 cells (Table 3). The latter was in contrast to hSK-Mel-28 and hM980409 cells expressing αvβ8, where the anti-β8 antibody and the FK-M1 and FK-M3 proteins reduced infection to the same levels as in hM000216 cells (Table 3). Possibly, some mouse cells express HSPG which M1 may use under certain instances [43,44].

Of note, none of the blocking agents had any appreciable effect in A549 cells infected with the fiber-chimeric H5-ΔE3B-CG-FK-M1 and -FK-M3 viruses, whereas in A549-hβ6 cells, the FK-M1 and FK-M3 proteins blocked infection by ~90%, and the anti-β6 antibody by ~60 to 70% (Table 3). M3-IX-G infection of A549-hβ6 cells was efficiently blocked by all three

**Table 3. Efficiencies of integrin Ab- and FK-mediated reduction of MAdV and chimeric HAdV reporter infection at 800 ng/ml.**

| Cell type / MFI of staining with indicated control and integrin antibodies | Blocking Agent | M1-IX-G | H5-ΔE3B-CG-FK-M1 | M3-IX-G | H5-ΔE3B-CG-FK-M3 |
|---|---|---|---|---|---|
| | | % inhibition of virus-mediated GFP expression | | | |
| CMT93 mouse colon cancer / con Ab: 304 anti-β6: 4,599 anti-β8: 357 | Ab-β6 | 84 | 85 | 100 | 86 |
| | Ab β8 | 9 | 0 | 4 | 0 |
| | FK-M1 | 100 | 99 | 100 | 98 |
| | FK-M3 | 100 | 99 | 100 | 98 |
| | FK-H5 | 0 | 0 | 0 | 18 |
| L929 mouse fibroblast / con Ab: 313 anti-β6: 710 anti-β8: 237 | Ab-β6 | 38 | 71 | 58 | 61 |
| | Ab β8 | 0 | 10 | 5 | 15 |
| | FK-M1 | 98 | 79 | 78 | 51 |
| | FK-M3 | 79 | 72 | 79 | 53 |
| | FK-H5 | 1 | 0 | 15 | 0 |
| 3T6 mouse fibroblast / con Ab: 229 anti-β6: 200 anti-β8: 1,483 | Ab-β6 | 0 | 0 | 0 | 0 |
| | Ab β8 | 70 | 62 | 92 | 60 |
| | FK-M1 | 61 | 61 | 60 | 55 |
| | FK-M3 | 52 | 52 | 85 | 52 |
| | FK-H5 | 0 | 0 | 0 | 0 |
| M000216 human melanoma / con Ab: 219 anti-β6: 259 anti-β8: 9,052 | Ab-β6 | 13 | 0 | 5 | 0 |
| | Ab β8 | 100 | 83 | 100 | 85 |
| | FK-M1 | 79 | ND | 97 | ND |
| | FK-M3 | 98 | ND | 99 | ND |
| | FK-H5 | 2 | ND | 0 | ND |
| SK-Mel-28 human melanoma / con Ab: 349 anti-β6: 324 anti-β8: 4,447 | Ab-β6 | ND | 1 | ND | 1 |
| | Ab β8 | 100 | 97 | 94 | 95 |
| | FK-M1 | ND[a] | 79 | ND | 82 |
| | FK-M3 | ND | 95 | ND | 92 |
| | FK-H5 | ND | 0 | ND | 6 |
| M980409 human melanoma / con Ab: 158 anti-β6: 173 anti-β8: 7,245 | Ab-β6 | 0 | 0 | ND | 0 |
| | Ab β8 | 96 | 70 | ND | 56 |
| | FK-M1 | 64 | 22 | ND | 23 |
| | FK-M3 | 95 | 66 | ND | 56 |
| | FK-H5 | 0 | 2 | ND | 0 |
| A549 human lung carcinoma / con Ab: 166 anti-β6:186 anti-β8: 201 | Ab-β6 | ND | 0 | ND | 7 |
| | Ab β8 | ND | 0 | ND | 6 |
| | FK-M1 | ND | 11 | ND | 10 |
| | FK-M3 | ND | 0 | ND | 10 |
| | FK-H5 | ND | 0 | ND | 0 |
| A549-hβ6 con Ab: 219 anti-β6:1,349 anti-β8: 210 | Ab-β6 | ND | 59 | 97 | 69 |
| | Ab β8 | ND | 6 | 0 | 0 |
| | FK-M1 | ND | 92 | 100 | 90 |
| | FK-M3 | ND | 91 | 99 | 88 |
| | FK-H5 | ND | 7 | 0 | 0 |

[a] ND, not determined.

reagents by >97% (Table 3). In A549 cells lacking or expressing moderate levels of αvβ6/8, the human fiber-chimeric viruses may bind to other integrins, such as αvβ5 by penton base. This possibility is based on the earlier notion that H5 infection of CAR-negative cells was

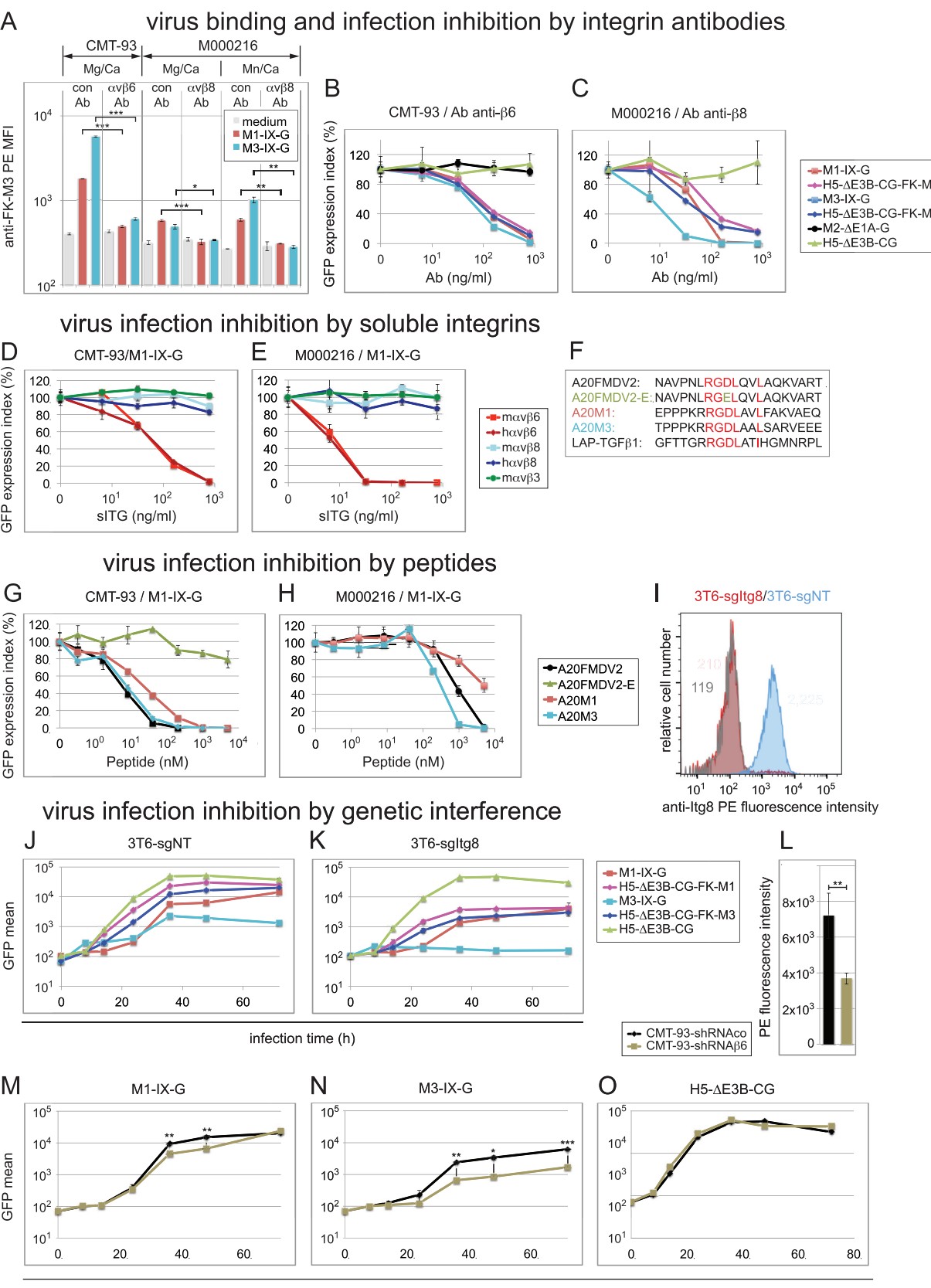

**Fig 6. Inhibition of virus binding and infection by β6- and β8-specific antibodies, sITGs, peptides and genetic interference.** (A) Virus binding interference in CMT-93 and M000216 cells by β6-/β8 function blocking antibodies. Detached cells were sequentially incubated for 1 h on ice with control antibody, or the anti-β6/-β8 antibodies, followed by incubation with control medium or medium containing the indicated viruses at an MOI of 4, the rabbit anti-FK-M3 antibodies, and finally the secondary PE-conjugate antibodies. Incubation and washing buffer contained either $Mg^{2+}/Ca^{2+}$, 1 mM each, or 1/0.2 mM $Mn^{2+}/Ca^{2+}$. (B-C) Virus infection interference in CMT-93 and M000216 cells by β6- and β8-specific antibodies. CMT-93 (B) and M000216 cells (C) were pre-incubated for 1 h on ice using 5-fold dilution series of the specific β6- or β8-integrin antibodies, respectively, starting with 800 ng/ml as highest concentration, followed by addition of the different indicated GFP-expressing viruses and transfer to 37˚C for 48 h. An MOI of 1 was used for CMT-93 cells and MOI of 3 for M000216 cells in all experiments shown in this figure. GFP analysis was performed 48 h pi, and expression index was normalized to a control antibody. $IC_{50}$ values determined in this experiment are summarized in Table 2. (D, E) Infection blocking assays by sITGs. M1-IX-G virus was incubated for 1 h at RT with 5-fold serial dilutions of the indicated sITGs starting from 800 ng/ml to 6.4 ng/ml, followed by addition to CMT-93 cells (D) and M00216 cells (E) and cultivated and further processed as above. (F-H) Infection blocking assays by peptides. (F) The 20-mer peptides tested for virus infection inhibition included peptides A20FMDV2 derived from the VP1 coat protein of FMDV2, A20FMDV2-E containing a D to E mutation in the critical RGD motif, A20M1 and A20M3 derived from M1-/M3-FK, respectively, as compared to LAP-hTGFβ1, all containing the critical αvβ6/αvβ8-binding RGDLXX(L/I) motif. (G, H) Cells were pre-incubated on ice with 5-fold serial dilutions of peptides resulting in final concentrations from 5,000 to 0.32 nM. Subsequently, M1-IX-G virus was added to CMT-93 cells (G) or M000216 cells (H), followed by processing as described above. (I) Comparative flow cytometry profiles of αvβ8 expression in 3T6 cells. Blue and red show β8-specific staining in 3T6-sgNT and 3T6-sgItgβ8 cells, respectively, and grey histogram shows background staining of 3T6-sgItgβ8 cells using a matched isotype control. Numbers indicate MFI values of specific or control antibodies. (J, K) Transduction of 3T6-sgNT and 3T6-sgItgβ8 cells using M1-/M3-IX-G, the fiber-chimeric H5-ΔE3B-CG-FK-M1/-FK-M3 and control H5-ΔE3B-CG at an MOI of 9. Cells were processed as described in Fig 1B. (L) Comparative flow cytometry MFI αvβ6 expression values in control CMT-93-sgNT versus β6 integrin shRNA knock down CMT-93-sgItgβ6 cells. (M-O) Infection of control CMT-93-sgNT and CMT-93-sgItgβ6 cells using M1-IX-G (M), M3-IX-G (N) and H5-ΔE3B-CG (O) at an MOI of 1. Cells were processed as described above. Except for the representative flow cytometry histogram in (I), data represent triplicates, shown as mean ± SEM. Asterisks indicate level of significance for comparison of indicated values ([*], $P < 0.05$; [**], $P < 0.005$; [***], $P < 0.0005$).

dependent on the RGD in penton base and efficiently blocked by antibodies to αvβ5 [88–90]. In conclusion, anti-integrin β6 and β8 antibodies potently block M1 and M3 cell binding and infections of mouse and human cells.

## Soluble integrin mouse αvβ6 blocks M1 and M3 infections

We found that human and mouse soluble integrin (sITG) αvβ6 blocked M1-IX-G infection of CMT-93 cells (MOI 1) at $IC_{50}$ of 80 and 83 ng/ml, respectively, whereas control sITG mαvβ3 had no effects (Fig 6D and Table 2). M3-IX-G gave similar results (S9A Fig and Table 2), akin to M1-IX-G and M3-IX-G in M000216 cells (Figs 6E and S9B and Table 2). In contrast, and for reasons unknown, mouse and human sITG αvβ8 did not affect M1 and M3 infections even if incubated at 37˚C instead of 4˚C, or in the presence of 1 mM $Mg^{2+}$, $Mn^{2+}$ and $Ca^{2+}$ prior to infection or 15 μM during the 48 h infection period (data are deposited at Mendeley Data). Notably, soluble mouse and human αvβ8 were chromatographically intact, as indicated by PAGE-blue analyses (S10 Fig), and an alternative source of soluble human αvβ8 also failed to block M3-IX-G infection of M000216 cells (S9B Fig).

## RGDLXXL containing peptides of FK-M1 and FK-M3 block M1 and M3 infections

We finally tested different 20-mer integrin ligand peptides in M1- and M3-IX-G infections (sequence information see Fig 6F). We included the A20FMDV2 peptide derived from the VP1 coat protein of foot and mouth disease virus 2 (FMDV2) [91]. This peptide has a reported $IC_{50}$ of 0.93 nM and 45 nM for inhibiting the binding of sITG αvβ6 and αvβ8 to the latency-associated protein component of transforming growth factor β substrate (LAP-TGFβ) [92]. As a control, we used the peptide A20FMDV2-E with a D to E substitution in the RGD motif. The third and fourth peptides were A20M1 and A20M3 derived from FK-M1 and FK-M3, respectively, containing as A20FMDV2 the critical αvβ6/αvβ8-binding RGDLXXL motif [93,94].

The A20FMDV2, A20M1 and A20M3 peptides gave low nanomolar inhibition profiles with M1-IX-G infections of CMT-93 cells (MOI 1), while the A20FMDV2-E peptide had only

a very weak effect at the highest concentration tested (Fig 6G and Table 2). Similar results were obtained with M3-IX-G ($IC_{50}$ values of 1 to 5 nM, and ~3,770 nM for A20FMDV2-E, see S11A Fig and Table 2), as well as the fiber chimeric H5-ΔE3B-CG-FK-M1 viruses ($IC_{50}$ values of 5 to 30 and > 5,000 nM for A20FMDV2-E, see S11B Fig and Table 2). No appreciable effects were seen in CMT-93 infections with M2-ΔE1-G or H5-ΔE3B-CG (S11C and S11D Fig and Table 2), indicating lack of toxicity. In contrast, the peptides A20FMDV2, A20M1 and A20M3 had only micromolar efficacy in M000216 cells infected with M1-IX-G at MOI 3 ($IC_{50}$ values of 422 to ~5,000 nM, see Fig 6H and Table 2), and intermediate effects on infection with M3-IX-G ($IC_{50}$ values of 17.9 to 165 nM, see S11E Fig and Table 2).

Collectively, A20FMDV2 and A20M3 peptides were about 5-10-fold more potent than A20M1 in blocking M1- and M3-IX-G transduction. The only exception was M3-IX-G transduction of β8-expressing M000216, where A20M3 was about 10-fold more potent than A20FMDV2 and A20M1. In summary, three distinct protein interference procedures, namely antibodies, sITGs, FKs and peptides derived thereof confirmed the important role of αvβ6 and αvβ8 in M1 or M3 infections.

## Inhibition of M1 and M3 infections by genetic interference with β6 and β8 integrins

We used CRISPR/Cas9-mediated knock-out (KO) to eliminate β8 expression in 3T6 cells, and shRNA-mediated knockdown of β6 in CMT-93 cells. The latter was necessary because four different CRISPR/Cas9 guide RNAs were unsuccessful to generate β6-KO CMT-93 cells. β8 expression reduction in 3T6-sg-Itgβ8 cells was 96% (Fig 6I), and β6 expression in CMT-93-shRNAβ6 cells was reduced by 50% (Fig 6L). The 3T6-sgItgβ8 cells were significantly less infected with M1- and M3-IX-G and the fiber-chimeric viruses (range ~4- to 13-fold reduction); especially M3 was strongly inhibited while the CAR-binding H5-ΔE3B-CG was not affected (Fig 6J and 6K). The CMT-93-shRNAβ6 knock-down cells were significantly less infected with M1-IX-G and M3-IX-G (2- and 3.7-fold reductions) compared to CMT-93-shRNAco control cells, while H5-ΔE3B-CG infection was not affected, indicating that the shRNA expression in these cells had no adverse effects on β6 independent infections (Fig 6M–6O). These results further underlined the importance of αvβ6 and αvβ8 for M1 and M3 infections.

## Surface plasmon resonance reveals high affinity, saturable binding of FK-M1 and FK-M3 to αvβ6 and αvβ8 integrins

To test if FK-M1 and FK-M3 directly bound to mouse and human αvβ6/αvβ8 integrins, we used surface plasmon resonance (SPR) and determined the interaction strength. FK-M1 and FK-M3 were biotinylated such that they retained their full inhibition power against M1 and M3 infection (S4C and S4D Fig), were immobilized and perfused with mouse or human sITG αvβ6/αvβ8. Low FK immobilization density on the sensor chip was generally sufficient to obtain good quality sensorgrams of sITG αvβ6 binding, while for sITG αvβ8, 10- to 20-fold higher FK immobilization densities were required. Sensorgrams were fitted with a 1:1 kinetic model, and kinetic data are summarized in Table 4. In the absence of $Ca^{2+}$, $Mg^{2+}$ and $Mn^{2+}$, no interaction of integrins with FKs was detected. In presence of divalent cations, the $K_D$ values for FK-M1 binding to mouse and human αvβ6 were in the low nM range (0.25 and 1.58 nM, respectively), and for FK-M3 in the pM range (< 10 pM and 59 pM, respectively) (Figs 7A, 7B, S12A and S12B and Table 4). FK-H5 did not bind to αvβ6 or αvβ8 (data are deposited at Mendeley Data). Human sITG αvβ8 gave $K_D$ values of 14.5 and 1.32 nM for FK-M1 and FK-M3 binding, respectively (S12C and S12D Fig and Table 4), whereas binding of mouse

**Table 4. SPR kinetic data of soluble integrin binding to immobilized FK-M1 and FK-M3.**

| FK Probe | Analyte | Surface density (response units) (RU) | $k_{on}$ ($10^4$ $M^{-1}s^{-1}$) | $k_{off}$ ($s^{-1}$) | $K_D$ (nM) | Chi$^2$ (RU$^2$) |
|---|---|---|---|---|---|---|
| M1 | mαvβ6 | 130 | 6.64 | $1.66 \times 10^{-5}$ | 0.250 [a] | 0.01 |
| M3 | mαvβ6 | 160 | 5.13 | $3.12 \times 10^{-8}$ | < 0.01 [b] | 0.29 |
| M1 | hαvβ6 | 130 | 8.02 | $1.27 \times 10^{-4}$ | 1.58 [c] | 0.01 |
| M3 | hαvβ6 | 160 | 8.50 | $5.01 \times 10^{-6}$ | 0.059 [d] | 0.02 |
| M1 | hαvβ8 | 3130 | 46.5 | $6.73 \times 10^{-3}$ | 14.5 [e] | 0.02 |
| M3 | hαvβ8 | 1340 | 72.9 | $9.65 \times 10^{-4}$ | 1.32 [f] | 0.26 |

[a] data corresponds to experiment shown in Fig 7A

[b] data corresponds to experiment shown in Fig 7B

[c] data corresponds to experiment shown in S12A Fig

[d] data corresponds to experiment shown in S12B Fig

[e] data corresponds to experiment shown in S12C Fig

[f] data corresponds to experiment shown in S12D Fig

sITG αvβ8 could not be measured, most likely due to chemical additives in the integrin sample.

## Biphasic high affinity binding of FK-M1 and FK-M3 to αvβ6 and αvβ8 expressing cells

Native integrins have high- and low-affinity states, representing extended-open and bent-closed/extended-closed conformations [50–52]. To probe for the native environment of αvβ6 and αvβ8 integrins FK-M1 and FK-M3 were incubated with cells containing defined αvβ6/αvβ8 expression levels followed by staining with anti-FK-M3 antibody as described in the virion-cell binding experiments (see Fig 5B and 5C). FK-M1 binding to B16-mβ6, as well as FK-M3 binding to B16-mβ6, B16-mβ8, CMT-93 (expressing αvβ6) and M000216 (expressing αvβ8) cells was biphasic and reached a plateau at low nM concentrations (Figs 7C–7E, S13A and S13B). At this concentration, there was no binding to parental B16 cells or EDTA-treated CMT-93 and M000216 cells, indicating high-affinity binding to αvβ6/αvβ8. Above the plateau, these FKs increasingly bound to parental B16 and EDTA-treated M000216 cells, indicating low-affinity attachment to sites other than αvβ6/αvβ8. Binding constants on high-affinity binding sites were derived from Scatchard plot analyses of subtracted binding data, revealing equilibrium dissociation constants ($K_D$) in the high-affinity range (examples shown in S14 Fig and summary of all experiments in Table 5). The average $K_D$ for FK-M1 binding to B16-mβ6 cells was 1.27 nM (an example is shown in S14A and S14B Fig), the average values for FK-M3 binding to CMT-93 and B16-mβ6 cells were 69 and 68 pM, respectively (example in S14C–S14F Fig), and finally the values for binding of FK-M3 to B16-mβ8 and M000216 cells were 1.31 and 1.21 nM, respectively, (example in S14G–S14J Fig).

Together, the SPR and the cell-based affinity measurements revealed $K_D$ values in the subna-nomolar and nanomolar ranges, that is FK-M1 values were about 10- to 60-fold lower for αvβ6 than for αvβ8, and the FK-M3 binding concentrations were about 20- to 130-fold lower for mαvβ6 than αvβ8. This suggests that αvβ6 is a more efficient receptor for M1 and M3 than αvβ8.

## Molecular modeling of FK-M1 and FK-M3 RGDLXXL peptides in complex with mαvβ6 and αvβ8 integrins

To explore the interaction landscape between the viral FKs and mouse αvβ6/αvβ8 integrins we modeled the ternary complex of two mouse integrin chains docked to the A20M1, A20M3 and

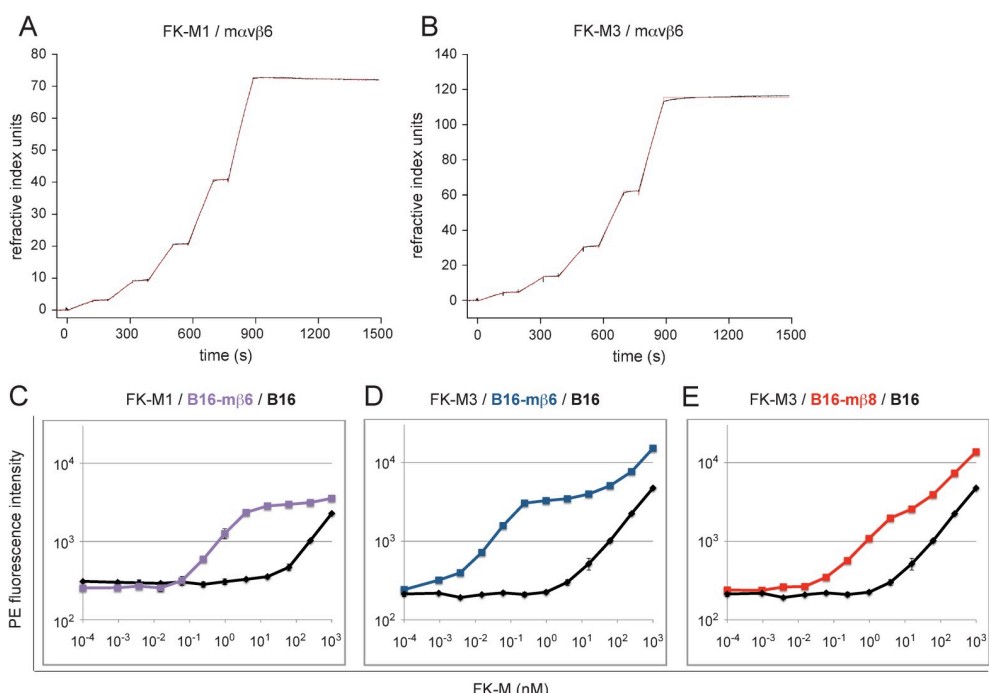

**Fig 7. FK affinity determination to sITG αvβ6 and αvβ8 by SPR analysis and to cell surface integrins by saturation binding.** (A, B) Sensor chips containing immobilized biotinylated FK-M1 and FK-M3 were probed with mouse sITG αvβ6. Following consecutive analyte injections over 120 s, dissociation was monitored for 600 s (black). Sensorgrams were fitted with a 1:1 kinetic model (red). (C-E) FK saturation cell binding assays using cells with defined αvβ6/αvβ8 expression included FK-M1 binding to B16-mβ6 (C), FK-M3 binding to B16-mβ6 (D) and FK-M3 binding to B16-mβ8 (E). Parental B16 cells were included in order to subtract background levels when calculating equilibrium dissociation constant $K_D$ values by Scatchard plot analyses.

A20FMDV2 peptides (S1–S6 Datasets), using the co-structures of αvβ6 and αvβ8 with TGF-β1 as a basis (see Materials and Methods). TGF-β1 harbors a similar amino acid sequence around the RGD motif as FK-M1, FK-M3 and FMDV VP1 (Fig 6F). A20M3 binds at the interface between the αv- and β6 subunits, where its Arg7 (equivalent to Arg215 of TGF-β1) is in an ionic interaction with Asp215 of αv, and Asp9 (equivalent to Asp217 of TGF-β1) binds the Mn²⁺ ion (Fig 8A and 8B). Gly8 of the RGD motif in FK-M1 and FK-M3 accommodates the bending of the peptide thereby allowing the side chain of Leu13 to dip into a hydrophobic pocket of the β6 and β8 subunits (Fig 8B and S1 and S2 Datasets). Thus, the FK-M1 and FK-M3-derived peptides fit perfectly well into the αvβ6 and αvβ8 binding sites. Notwithstanding that the crucial interactions of the RGDLXX(L/I) motif are conserved, differences exist at

**Table 5. Equilibrium data of high-affinity knob interaction measured by binding saturation to cellular integrins.**

| cells | $K_D{}^a$ to FK-M1 (nM) | $K_D{}^a$ to FK-M3 (nM) |
|---|---|---|
| CMT-93 | | 0.069 [a] |
| B16-mβ6 | 1.27 [b] ± 0.66 | 0.068 [b] ± 0.018 |
| B16-mβ8 | | 1.31 [b] ± 0.29 |
| M000216 | | 1.21 [c] |

[a] represents 2 triplicate measurements

[b] represents 3 triplicate measurements

[c] represents single triplicate measurement

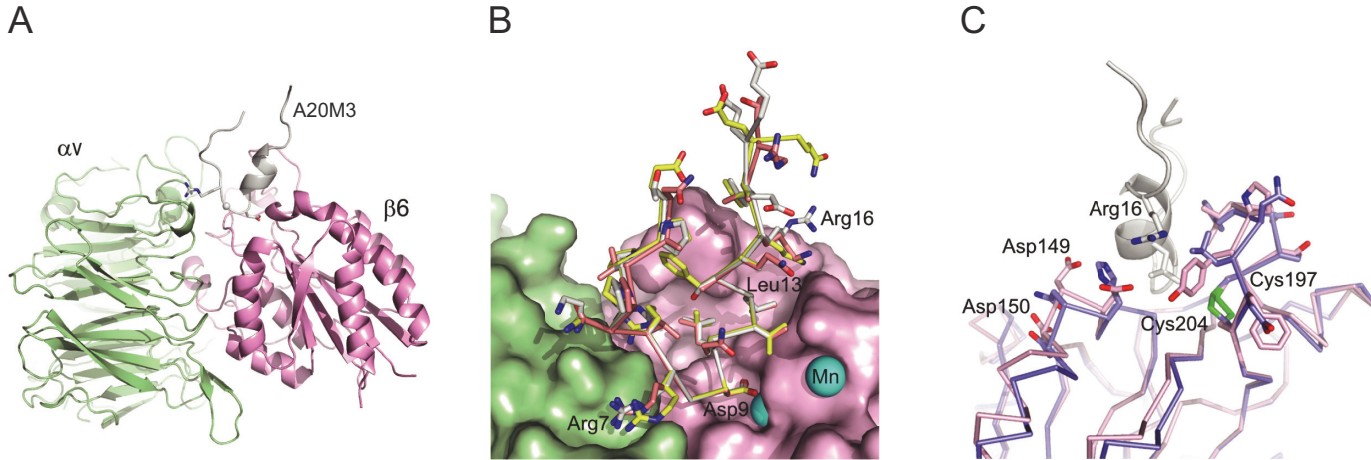

**Fig 8. Molecular docking simulations of FK-M1 and FK-M3 RGD-containing peptides with mouse αvβ6 and αvβ8 integrins.** (A) Overview of αvβ6 complexed with the 20mer A20M3 peptide. αv (light green), β6 (pink) and A20M3 (grey) chains are shown as cartoon traces. Side chains of the RGD motif are shown as sticks and spheres. (B) Superposition of αvβ6 in surface representation in complex with A20M1 (yellow carbons), A20M3 (grey carbons) and A20FMDV2 (salmon carbons). The position of manganese ions (cyan spheres) shown superimposed were derived from the human αvβ6/TGFβ1 complex (PDB ID: 5ffo). Labeled aa refer to the A20M3 sequence. (C) Superposition of the αvβ6/A20M3 complex onto the αvβ8 structure (blue carbons). Note that the A20M3 peptide is not shown here. Residue numbering refers to the Uniprot entry Q9Z0T9 (mouse integrin subunit β6) and the A20M3 peptide sequence (Arg16).

the periphery of the peptide binding sites. This is emphasized by superposition of the mouse αvβ6/A20M3 and αvβ8/A20M3 models (Fig 8C). The loop formed by the β6 residues 197–204 and the β8 residues 211–218 opposes against the α-helix of the A20M3 peptide. The β6 and β8 loops comprise completely different aa sequences, yet both of them are constrained by a conserved disulfide bridge and can accommodate the side chain of Leu13 from the RGDLXX(L/I) motif. Furthermore, residues 146–148 located on the surface of β6 form a cluster of negative charges in close proximity to Arg16 of the A20M3 peptide, allowing a charge-charge interaction between αvβ6 and A20M3. This cluster is absent in the mouse β8 integrin, and hence might contribute to a higher affinity of FK-M3 to αvβ6 than to αvβ8. Notably, both the disulfide constrained loops of mouse β6 and β8 and the negatively charged patch of β6 are present in the structures of human αvβ6 and αvβ8 integrins (PDB IDs: 5ffo and 6uja). Together, the modeling data reinforce the notion that the RGD motif and adjacent sequences of FK-M1 and FK-M3 bind with high affinity directly to αvβ6 and with somewhat lower affinity to αvβ8 present on both mouse and human cells. These binding interactions may provide a basis for potential cross-species transmission of M1 and M3.

## Discussion

Integrins connect the extracellular matrix with the cytoskeleton. They are used by nonenveloped and enveloped viruses for attachment and cell entry [95,96], and by bacteria for adhesion and translocation of virulence factors [97]. In most cases integrins recognize protein motifs, for example an RGD sequence evolved in viral envelopes or capsids. Here, we identified the RGD-binding integrins αvβ6 and αvβ8 as high affinity receptors for the FKs of M1 and M3, as based on six criteria. (i) Integrin profiling of M1 and M3 resistant and permissive mouse and human cells showed that αvβ6/αvβ8 expression correlates with virus infection. (ii) Nonpermissive B16 cells become permissive for M1 and M3 infections upon transfection of integrin β6 or β8 subunits. (iii) Specific anti-αvβ6 or anti-αvβ8 integrin antibodies block M1 and M3 binding and infections. (iv) Soluble integrin αvβ6 and synthetic 20-mer peptides containing the high

affinity αvβ6/αvβ8 binding motif RGDLXXL block M1 and M3 infections. (v) M1 and M3 infections are attenuated by CRISPR/Cas9-mediated knock-out of αvβ8 in 3T6 cells, and by shRNA-mediated knock-down of αvβ6 in CMT-93 cells. (vi) SPR and cell-based assays reveal high affinity, and biphasic saturation binding of FK-M1 and FK-M3 to mouse and human αvβ6/αvβ8 integrins. Importantly, unlike penton base of H2 or H5, FK-M1 and FK-M3 harbor an elaborate receptor-binding element with two adjacent motifs, RGD and LXXL. This two-component element allows for tunable binding, at high affinity to extended-open (active) integrins, and somewhat lower affinity to closed (inactive) integrins [98]. This feature may allow the M1 and M3 virions to undergo drifting motions on inactive integrins dynamically engaged with the cytoskeleton, and thereby trigger their uncoating program for infection.

Eight RGD-binding integrins are known for their distinct RGD ligand preference [55,94,99]. αvβ6 and αvβ8 stand out for high affinity binding to a small number of non-integrin ligands, including LAP-TGF-β1 and LAP-TGF-β3 [82,84,100,101]. They are potent activators of TGF-β1/-β3 by inducing allosteric changes in LAP-TGF-β (in case of αvβ6) or activating matrix metalloproteinase-14, which proteolytically releases TGF-β. Mice lacking αvβ6 and αvβ8 have abnormalities similar to TGF-β1 and TGF-β3 null animals [102].

Other viruses than M1 and M3 evolved to use αvβ6 and αvβ8 as receptors. The VP1 protein of FMDV, a member of the *Aphthovirus* genus of the family *Picornaviridae* contains the αvβ6/αvβ8 consensus binding motif RGDLXXL, and binds and infects cells expressing αvβ6 [103] and αvβ8 [104]. In addition, other *Picornaviridae* family members, such as human parechoviruses also use αvβ6 as receptor, and Coxsackievirus A9 (CAV9) harbors a RGD(M/L)XXL motif in VP1 for high affinity binding to soluble αvβ6 receptor [105–107]. Enveloped viruses using αvβ6/αvβ8 integrins for cell entry include herpes simplex virus [108], and Epstein-Barr virus [109], possibly through an RGDNVAT sequence of HSV-1 gH and KGDEHVL of EBV gH.

We demonstrate that M1 and M3 evolved a dual αvβ6/αvβ8 integrin-binding motif, RGDLAVL in FK-M1 and RGDLAAL in FK-M3. Both the RGD and the LXXL motifs have been known from phage display studies to bind independently with high affinity to αvβ6 [93]. The presence of two directly adjacent integrin binding motifs is thought to yield very high-affinity TGF-β1 interaction with both αvβ6 and αvβ8 integrins [82–84,91,94]. Our structure-based modeling showed that the RGD-proximal helix of FK-M1 and FK-M3, in particular the RGD+4 aa, binds in a hydrophobic pocket on the integrin β6 and β8 surfaces. This well-known "specificity" loop has been described to be variable among integrin β chains and determines integrin ligand specificity [91,110]. The presence of two directly adjacent integrin binding motifs in FK-M1 and FK-M3 likely confers selective and very high affinity binding to αvβ6 and αvβ8 integrins, respectively. These interactions may be reinforced by avidity effects, as demonstrated for H3 and H7 binding to CD46 [74].

It will be interesting to test the function of the αvβ6 and αvβ8 integrins for M1 and M3 infection *in vivo*. M1 has tropism for endothelial cells, the monocyte/macrophage lineage, and astrocytes [34,39,111]. The highest viral loads are found in the brain, spleen, and spinal cord in susceptible mice, yet the virus rarely infects epithelial cells of the kidney [111], although kidney cells can become persistently infected leading to sustained viremia (for review see [28]). In addition, intranasal application of M1 leads to respiratory tract infections [30], and intraperitoneal or intranasal applications cause myocarditis accompanied by myocyte and endothelial necrosis, the latter possibly due to indirect systemic effects [31,112]. Possibly, peripheral blood monocytes or dendritic cell precursors provide the primary mechanism for virus dissemination during acute infection, which would be compatible with high levels of αvβ8 expression in monocytes, macrophages and dendritic cells [111]. Interestingly, side-by-side experimental infections of mice showed that organ tropism (measured by viral DNA levels) was markedly

different with M1 and M3 [23]. For example, M3 DNA was completely absent in the brain, but M1 DNA highly abundant. It is unknown at present, if this difference is due to differential abundance of cells expressing αvβ6 or αvβ8, or due to other brain cell-specific factors or immunity.

The identification of adenoviruses that bind with high affinity to αvβ6 and αvβ8 integrin receptors has implications for cancer research [7]. In the normal adult organism, αvβ6 is poorly expressed, with moderate levels in the epithelia of the kidney, uterus, bladder, respiratory tract, heart, intestine and salivary gland but not in the spleen, CNS or brain ([113] and NCBI Resources (https://www.ncbi.nlm.nih.gov/gene?Db=gene&Cmd=DetailsSearch&Term=16403). However, αvβ6 is upregulated in the context of tissue remodeling, wound healing, inflammation, and carcinogenesis [114]. In contrast, a wide variety of cell types constitutively expresses αvβ8, including kidney cells, epithelial cells, fibroblasts, and astrocytes, as well as monocytes, macrophages, dendritic cells and regulatory T cells [115] and NCBI Resources (https://www.ncbi.nlm.nih.gov/gene?Db = gene&Cmd = DetailsSearch&Term = 241226). αvβ6 integrin is overexpressed in more than one third of carcinomas, and is a biomarker for tumor cell invasiveness of gastric, pancreatic, cholangiocellular, breast, ovarian, colon, and head and neck cancer, with poor survival prognosis (for review see [49]). Because αvβ6 is largely absent in healthy tissues, this makes it an interesting target for therapies using monoclonal antibodies, peptides, oncolytic viruses, or chimeric antigen receptor-engineered T cells.

αvβ8 integrin is widely present in malignant cells of a variety of human carcinomas and melanomas but absent in tumor stroma and immune cells. This raises the possibility that αvβ8 expression in tumor cells is a driver of tumorigenesis mediated by TGF-β1 [115]. Accordingly, the antagonistic αvβ8 integrin antibody PF-06940434 is currently being tested in a clinical trial against various cancers, including melanoma and renal cell carcinoma, as well as ovarian, gastric, esophageal, and lung squamous tumors (NCT04152018). Likewise, neutralizing anti-β8 integrin antibodies or RGDLXX(L/I)-containing peptide mimetics have been suggested for treatment of fibrosis [116]. Interestingly, the A20FMDV2 peptide has been chemically altered into a drug conjugate for pre-clinical anti-cancer studies [117]. This peptide sequence was also genetically integrated into the HI loop of FK-H5 and shown to enable efficient infection of αvβ6-positive cells in a CAR-independent manner [118,119]. Our fiber-chimeric H5-ΔE3B-CG-FK-M1/-FK-M3 viruses targeting αvβ6/αvβ8 integrins are a promising tool for cancer diagnostics, as these viruses are naturally blind for CAR, complement component C4-binding protein and highly prevalent FK-H5 neutralizing antibodies. These viruses can easily be modified, for example by detargeting the liver, or blinding for coagulation factors and αvβ3 and αvβ5 integrins [19,89,120–122]. They add to a growing list of medically promising agents targeting αvβ6/αvβ8 integrins in cancer therapy.

## Materials and methods

### Cells

Cells grown in DMEM (D6429, Sigma Aldrich) + 8% fetal bovine serum (FBS, S0115, lot 1107A, Biochrom AG, Berlin) included human embryonic retinoblast 911 and kidney 293T cell lines, human melanoma SK-Mel-28 and DX3 (R. Dummer, University Hospital Zurich Dermatological Clinic), human lung carcinoma A548 (ATCC), mouse rectal carcinoma CMT-93 (S. Compton, Yale University School of Medicine, USA), mouse fibroblast NIH 3T6 (K. Spindler, University of Michigan, Ann Arbor, USA), L929 (ATCC), pancreatic Panc02 (R. Alemany, Institut d'Investigació, Biomèdica de Bellvitge, Barcelona Spain), Chinese hamster cells CHO (ATCC), and mouse melanoma B16-F1 (ATCC). NMU-MG-Fucci (RIKEN, Japan) were grown with additional 10 µg/ml human insulin (I9278, Sigma Aldrich). Cells grown in

RPMI1640 (R8785, Sigma Aldrich) + 8% fetal bovine serum included primary short time melanoma cultures M000216 and M980409 [65].

B16-mCAR cells stably expressing a N-terminal FLAG-tagged mCAR were generated by transfection of B16-F1 cells with pcDNA3.1-turbo-neo-mCAR1-FLAG, followed by selection in medium containing 2 mg/ml G418 (PAA Laboratories) and fluorescence-activated cell sorting of pooled positive cell clones stained with the anti-FLAG antibody. For generation of pcDNA3.1-turbo-neo-mCAR1-FLAG, a PCR introducing the FLAG-tag between the mCAR signal peptide and the mature mCAR sequence was performed. We used the pCMVsport2m-CAR1 plasmid as a template (J. Bergelson, Pereleman School of Medicine, Philadelphia, Pennsylvania). Using HindIII/XbaI restriction sites, the fragment was cloned into a pcDNA3.1-turbo-neo vector, containing a CMV promoter and a neomycin resistance cassette [79].

B16-mβ6 and B16-mβ8 cells were generated by stably transfecting B16 cells with pCMV3-mITGB6 (Sino Biological, MG50097-UT) and pcDNA3.1-mβ8ITG-HB (kindly provided by D. Sheppard, UCSF), respectively, followed by selection in medium containing hygromycin B at 0.4 mg/ml. A549-hβ6 cells were generated by stably transfecting A549 cells using pcDNA1-neo-huITGβ6 (Addgene, Watertown MA, USA) followed by selection in medium containing neomycin at 1 mg/ml. Clonal cells were obtained by limiting dilution cloning.

Generation of pooled 3T6-sgItgb8 knockout cells was performed as described previously using a lentiviral vector containing the single guide RNAs (sgRNAs) and a high-fidelity Cas9 (Lenti-eCas9) [123]. The sequences used as gRNA templates are indicated in S2 Table. gRNAs were cloned into Lenti-eCas9 according to the instructions of the Zhang lab. All plasmids were verified by Sanger sequencing. The plasmids were then used for the generation of lentiviral particles. For the production of lentivirus, $4.5 \times 10^6$ 293T cells were seeded on a 10 cm dish. The next day, the cells were transfected with 3.4 μg pVSV-G (Clontech), 6.5 μg pCMV-dR8.91-Gag-Pol, and 10 μg of lentiviral expression plasmid using the calcium phosphate method. The culture supernatant was changed to fresh medium the following morning, and the supernatant containing the lentiviral particles was harvested at 2 days post transfection and filtered through a 0.45 μm filter before storage at -80˚C. 3T6 cells were transduced with lentiviral supernatant containing Itgb8-targeting gRNAs. After two days, the cells were selected with 2 μg/ml puromycin until all non-transduced cells were dead (ca. 5 days). Surviving cells from wells in which at least 50% of the cells had died were then expanded and frozen. Pooled stable RNA interference knockdown CMT-93-shRNAβ6 and -shRNA control cells (CMT-93-shRNAcon) were generated by transduction with lentiviral particles sc-43136-V and sc-108080 (Santa Cruz Biotechnology) followed by selection in medium containing 2 μg/ml puromycin. CHO-15B6 and the thereof derived CHO-hCD46 have been described elsewhere [74]. CHO-hDSG-2 was generated using lentiviral particles (pLenti-Ef1a-DSG2, kindly provided by A. Lieber, University of Washington, Seattle) for transduction of CHO cells followed by cell sorting of DSG-2-positive cells. Generation of mouse CMT-93, 3T6 and NMU-MG-Fucci cells stably expressing I-SceI endonuclease for rescue of infectious virus from circular pKSB2-BAC-mids was accomplished following transduction with MLV-ER-I-SceI-HA [124,125]. Selection for resistant cells was done in medium containing 3 μg/ml puromycin for CMT-93 and 3T6 cells, and 2 μg/ml for NMU-MG-Fucci cells. In these cells, nuclear translocation of ER-I-S-ceI-HA can be induced upon treatment with 1.5 μM 4-OH-tamoxifen.

## Viruses and virus progeny assay

Wt virus batches of M1 (K. Spindler, University of Michigan, Ann Arbor, USA), M2 (S. Compton, Yale University School of Medicine, USA), and M3 (DH Krüger, Chrité Campus Mitte,

Berlin Germany) were mainly amplified in CMT-93 cells. H5 wt (wt300) (P. Hearing, Stony Brook University, New York, USA), and H35 (ATCC) were amplified in A549 cells. Viral genomic DNA was isolated using a standard phenol/chloroform method, which included extended proteinase K treatment.

Generation of the nine novel recombinant virus constructs (Table 1 and S1 and S2 Figs) is described in detail in S2 Text. MAdV and HAdV gene region annotations were taken from (https://sites.google.com/site/adenoseq/) maintained by B. Harrach (Veterinary Medical Research Institute, Hungarian Academy of Sciences, Budapest, Hungary). In the HAdV-C5-derived vectors H5-ΔE3B-CG-FK-M1 and H5-ΔE3B-CG-FK-M3 the endogenous FK was replaced with the FK of M1 and M3, respectively. For this purpose, the H5 FK sequence encompassing aa residues 404 to 581 were replaced with residues 391 to 613 of M1 and 357 to 567 of M3, respectively. Recombinant HAdVs were plaque-purified in 911 cells, followed by amplification in either A549 or 911 cells, depending on the E1 status. The HAdV-derived vectors were concentrated and purified by standard double CsCl procedure, followed by titer determination using plaque assays [65]. Recombinant MAdVs were mostly amplified in CMT-93 cells, except M2-ΔE1A-G and M3-ΔE1A-G, which were amplified in NMU-MG-Fucci cells. Amplification in CMT-93 cells gave rise to an evident cytopathic effect after 3–7 days and supernatants were cleared by centrifugation for 20 min at 4,000 rpm and 4˚C. Supernatants were stored at 4˚C or kept at -20˚C. When amplified in NMU-MG-Fucci cells, CPE was less evident, and supernatants were regularly harvested and tested for virus presence. For all experiments, recombinant MAdVs were used as unpurified supernatant virus. This was done for two reasons. First, MAdVs are efficiently released into the supernatant of infected cells [69,126], and second, we observed that in our hands, standard CsCl purification led to considerable drop of infectivity of these viruses. Recombinant MAdV stocks were quantified by cytofluorometric GFP assays to determine infectious GFP titers. For this, three-fold dilution series of the virus solutions were prepared and used to infect $5x10^4$ CMT-93 cells for 48 h. GFP positive cells were determined by flow cytometry, and the infectious titer of the viral stock was calculated using the Poisson distribution [127]. For infectious titer determination of MAdV wt, the same procedure was used but staining of cells was achieved by the use of mouse monoclonal anti-AdV pan-hexon antibody 8C4 (10R-A115c, Fitzgerald Industries) in permeabilized and fixed infected cells in combination with a secondary PE-anti mouse IgG antibody. Titers are summarized in Table 1.

For virus progeny assays, CMT-93, B16, B16-mβ6 and B16-mβ8 cells were infected in triplicates with M1-/M3-IX-G using an MOI of 1.5. After 14 h, the cells were thoroughly washed, trypsinized and re-seeded. Virus-containing supernatant samples were collected 48 and 72 h pi and used for titration analyses. Based on the virus input, fold increases of progeny virus were calculated.

## Generation of recombinant proteins

For production of recombinant E1A, E1B19K, protein IX and the different FK versions, the expression vector pET28a and pET20b-A(H)6-AviTag-GST (parental plasmids from Novagen) were used. The pET28a plasmid contains a His$_6$-tag N-terminally fused via a thrombin cleavage site to the inserted ORF. DNA fragments containing the ORFs were PCR-amplified using viral genomic DNA as template (except for E1A) and inserted into the corresponding cloning sites of pET28a. The oligonucleotide sequences (Microsynth AG, Switzerland) used for cloning are listed in S3 Table, together with protein sequences and relative molecular sizes of the natural and final his-tagged fragments. For the E1A constructs, cDNAs made from RNA of virus-infected CMT-93 cells were first PCR-amplified using a forward primer in combination with

an oligo-dT primer, followed by cloning into pBluescript. Sequence-confirmed plasmids were then used as templates. All N-terminal His$_6$-tagged proteins were purified by single-step nickel-nitrilotriacetic acid (Ni-NTA)-agarose affinity chromatography. The tag was not removed for the blocking experiments. Generation of FK-H5 was reported earlier [74].

For M2, three different versions of FK proteins were generated, including v1 (557–787), v2 (586–787) and v3 (517–787), the last two according to Singh et al. [75]. pET28a expression plasmids for the 586–787 and 517–787 forms were kindly provided by Mark J. van Raaij (Departamento de Estructura de Macromoléculas, Centro Nacional de Biotecnologia, Madrid, Spain), but we also generated our own version of pET28a-FK-M2v3.

In order to express biotinylated M1- (termed FKb-M1), M3- (FKb-M3) and FK-H5 (FKb-H5) proteins, the coding sequences were cloned into pET20b-containing N-terminal His$_6$- and Avi-tag [128]. To produce the biotinylated proteins, the host strains were co-transformed with plasmid pBirA (Avidity) expressing protein biotin ligase. The degree of biotinylation was tested using a streptavidin shift assay in combination with PAGE-blue analysis.

The molar mass, biophysical properties and oligomeric state of the different knob variants were analyzed using size exclusion chromatography (SEC; LC1100 System, Agilent Technologies, Santa Clara, CA) coupled to an Optilab rEX refractometer (Wyatt Technology, Santa Barbara, CA) and a TREOS three-angle light-scattering detector (Wyatt Technology). For protein separation, 50 μl of a protein solution at a concentration of 30–40 μM was injected onto a 24 ml Superdex 200 10/30 column (GE HealthcareBiosciences, Pittsburg, PA) at 0.5 ml/min in PBS (137 mM NaCl, 2.7 mM KCl, 10 mM Na$_2$HPO$_4$, 2 mM KH$_2$PO$_4$, pH 7.4). The determination of the molar mass was carried out with the ASTRA6 software (Wyatt Technology) using refractive index (RI) and light scattering signals.

Examples for the four mainly used FKs including FKb-M1, FK-M2-517-787, FK-M3 and FKb-H5 are shown in S5 Fig and summarized in S1 Table. For all four variants the mass recovery was >97%, indicating a low (or even no) tendency for insoluble aggregates. Expression levels from the FK-M1 expression system were noticed to be low, resulting in a generally poorer protein quality, e.g., in the FKb-M1 knob batch a mixture was found, consisting of almost 20% monomer at a molar mass of 31.8 kDa, and a main peak at 93.3 kDa, consistent with the desired trimeric knob. FK-M3 had a low tendency for the formation of trimer-dimers (5.8%), but the main peak's molar mass was determined as trimer with 90.8 kDa (theoretical molar mass: 92.7 kDa). For FK-M2-517-787 (theoretical molar mass of 99.9 kDa for the trimer), a single peak was observed with a mass recovery of 98% at a molar mass of 86.2 kDa, which, however, was somewhat smaller than the expected 99.9 kDa. The reason for this discrepancy remained unclear, but it is possibly that this represents a mixture of trimer and dimer. For FKb-H5, a single peak was observed at 71.3 kDa (expected molar mass of the trimer: 72.9 kDa) with a mass recovery of 100%.

## Transduction and virus blocking experiments

For GFP expression-based transduction analysis, triplicates of 10$^5$ cells were seeded in 12-well plates, incubated for 8 h at 37˚C in 5% CO$_2$, and transduced with recombinant GFP-expressing AdVs at various virus concentrations. Cells were harvested and processed for cytofluorometric analysis, including fixation with 3% paraformaldehyde, at various time points pi.

For competition experiments with FKs, antibodies, sITGs and peptides, cells were seeded in triplicate in 12-well plates as described above, and were allowed to adhere overnight. For competition with FK proteins, serial 5-fold dilutions of the different FKs were prepared in DMEM-8%/FBS, resulting in concentrations of 800, 160, 32, 6.4, 1.28 and 0.26 ng/ml (corresponding to 8.60, 1.72, 0.344, 0.0688, 0.0138, and 0.00275 nM). 500 μl of FK dilutions were added to the

cells and incubated for 1 h on ice under constant shaking. Similarly, for blocking with anti-integrin antibodies, dilutions from 800 to 6.4 ng/ml (corresponding to 5.33 to 0.0427 nM), and for peptide blocking assays, dilutions from 5,000 to 0.32 nM were applied.

Subsequently the different GFP-expressing viruses were added, using a MOI 1 or 3, and the cells were transferred to 37˚C for 48 h. For assays using the rabbit anti-FK neutralizing antibodies, serial 5-fold dilutions using dilution factors from 1:1,250 to 1:781,250, and for competition assays with sITGs serial 5-fold dilutions from 800 to 6.4 (or 1.28) ng/ml (corresponding to 3.17 to 0.0254, or 0.00508 nM) were prepared. These dilutions were subsequently incubated with viruses for 1 h at room temperature (RT), followed by addition to cells and processing as described above. Higher serum concentrations than 1:1,250 gave rise to cytotoxic effects, likely due to presence of thimerosal and glycerol added to these antibody batches [62] and data showing cytotoxic effects were omitted from the calculations. The data were normalized to the GFP expression values mediated by virus mixed with control anti CD4 antibody or FK-H3 (as a non-binding control) in PBS/glycerol. Mean values and standard deviations of triplicates from one representative experiment are shown. Statistical evaluation was performed using Student's $t$ test in Excel.

## Peptides, sITGs, antibodies

All four synthetic peptides used in this study (Fig 6F) were obtained from Psyclo Peptide Inc Shanghai, China, and were dissolved at 1 mg/ml in $H_2O$ and stored in aliquots at -80˚C. The 20-mer peptide A20FMDV2 derived from VP1 of FMDV2 (Uniprot: P03305 725–935) was described in [91], as was A20FMDV2-E, which contains a D to E mutation in the critical RGD motif. The A20M1 and A20M3 consist of aa residues 490–509 of the M1 fiber (Uniprot P19721), and 452–471 of the M3 fiber (GenBank: ACJ14524.1) sequence, respectively.

sITGs were obtained from R&D systems (Bio-techne brand, USA), including mouse and human αvβ3 (7889-AV-050/3050-AV-050), αvβ5 (7706-AV-050/2528-AV-050), αvβ6 (7480-AV-050/3817-AV-050), αvβ8 (8314-AV-050/4135-AV-050), recombinantly expressed in and secreted from CHO cells (quality control shown in S8 Fig). Human αvβ6 (IT6-H52E1) and αvβ8 (IT8-H52W4) were alternatively obtained from Acro Bisosystems (USA), recombinantly expressed in and secreted from 293 cells. Blocking activities of the different batches of αvβ6/αvβ8 were found to be interchangeable.

Polyclonal rabbit antibodies against the purified His-tagged E1A of M1, M2 and M3, E1B-19K of M1, and FK of M2 and M3, proteins produced in *E. coli* were generated following immunization of rabbits (Biogenes GmbH, Berlin, Germany). For this purpose, two animals were first immunized with 200 μg of purified protein using complete Freund's adjuvant, followed by three booster immunizations using 100 μg and incomplete Freund's adjuvant. Final bleeds were done following a fourth booster. The obtained sera were mixed with thimerosal at 0.02% as preservative. Production of rabbit anti-M1 and M2-hexon antibodies was described elsewhere [62]. Antibodies for Western included mouse monoclonal anti-tubulin antibody (DM1A, Sigma) and anti-GFP antibodies (clone 7.1 and 13.1, Roche). Secondary antibodies included HRP-conjugated donkey anti-rabbit IgG, and sheep anti-mouse IgG (GE Healthcare).

To detect FLAG-tag by flow cytometry, the M2 anti-FLAG F1804 (Sigma) was used. Human CD46 and DSG-2 were measured with antibodies MCI20.6 (gift from Denis Gerlier (Université Lyon, Lyon, France) and 6D8 (Santa Cruz Inc), respectively. Integrin antibodies included human αv mAb Sc-9969 (Santa Cruz), human β1 mAb sc-59829 (Santa Cruz), hu β3 mAb AP3 (ATCC), human β5 mAb B5-IVF2 (provided by M. Hemler, Harvard Medical School, Boston, USA), human αvβ3 mAb 23C6 (sc-7312, Santa Cruz), human and mouse αvβ5

mAb ALULA (provided by D. Sheppard, UCSF [129]), human and mouse αvβ6 mAb 10D5 (function blocking [86], ab77906, Abcam), human and mouse αvβ8 mAb ADWA-11 (function blocking, provided by D. Sheppard, UCSF [87]), mouse αv rAb RMV7 (14–00512, Affymetrix eBioscience), mouse β1 mAb (MAB2405, R&D systems, Bio-techne brand, USA) mouse β3 (rAb MAB41182, R&D systems, Bio-techne brand, USA). The mouse IgG1 isotype control included 9E10 anti-myc-tag antibody (Abcam). Secondary fluorescently labeled antibodies included goat anti-mouse (ref 103009), goat anti-rat (ref 303009), and donkey anti-rabbit (ref 644009), and these IgG-phycoerythrin antibodies which were purchased from Bio-Rad.

## Flow cytometry

For cytofluorometric analysis of surface proteins, cells were detached either by treatment with PBS-EDTA-20 mM, followed by washing and recovery in DMEM + 8% FBS for 90 min, or with trypsin (after confirming that trypsin did not destroy the antigens of interest on the cell surface). For integrin analyses and PBS-EDTA-20 mM-detachment, cells were washed and allowed to recover for 1 h as described by Rajan et al. [130]. Subsequently, $2x10^5$ cells were incubated with 1 μg of specific antibody, followed by incubation with 1 μg of secondary conjugate antibodies. Staining assays were performed in FACS buffer plus $Mg^{2+}/Ca^{2+}$ (PBS plus 2% FBS, $Mg^{2+}/Ca^{2+}$, 1 mM each, 0.02% sodium azide), since omission of $Mg^{2+}$ was reported to reduce staining signals by about 50% for one of the used antibodies (αvβ6 antibody ADWA-11 [131]). Flow cytometry analyses were performed on a FACSCanto II with FACSDIVA software (BD Biosciences) at the Flow Cytometry Facility of UZH Irchel. Data analysis was performed with FlowJo 10.4. In all experiments, MFI data represent the geometric mean of fluorescence intensities.

## PAGE and Western blot

For Western immunoblot analysis of viral proteins, cells infected at an MOI of 3 were lysed in NETN (10 mM Tris pH 8.0, 200 mM NaCl, 1 mM EDTA, 0.5% NP40) supplemented with protease inhibitors (Mini-Complete, Roche). Analyses of cleared cell lysates obtained from equal fractions of plate wells were performed by polyacrylamide gel electrophoresis [132] followed by Western blotting of electro-transferred protein to Immobilon-P membranes as described previously [80]. Membranes were saturated in TBS-T containing 5% dry milk and incubated with primary rabbit antibodies at 1:3,000. The mouse monoclonal antibodies were used at 1 μg/ml. Secondary HRP-conjugate antibodies were used at 1:4,000, for 1 h at RT. The immunoreactivity was determined using the Luminata Crescendo Western HRP substrate (Millipore) and scored using the ImageQuant LAS 4000 imager (GE Healthcare). Due to high background obtained with some of the novel rabbit antisera, Western blots sets were in part made from different gels. Whole blot data are deposited at Mendeley Data.

## Surface plasmon resonance analysis for kinetics/affinity between soluble integrins and FK-M1/FK-M3

Surface plasmon resonance (SPR) measurements were conducted using sensor chips SAD200M (Xantec, Düsseldorf, Germany) in a Biacore T200 instrument at 20˚C. For protocol development, other surfaces were tested including CM5 (Cytiva, Freiburg, Germany), CMD200M/ SAD200M, ZC30M/ ZC80M, and HLC200M (all Xantec). In the presence of divalent ions of $Ca^{2+}$, $Mg^{2+}$, and $Mn^{2+}$, tested integrin analytes tended to adsorb to all SPR chip surfaces. The nonspecific adsorption was reduced by addition of 0.1 mg/ml BSA and injection of very low integrin concentrations in the nM range. This approach was successful for the αvβ6 integrins but failed for αvβ8 because of low signal intensity that required a higher

immobilization density of FKs. Chips were coated with streptavidin following recommended protocols of the chip manufacturer. Immobilization of the biotinylated FKb-M1/-M3 and the negative control FKb-H5 were conducted in HBS buffer at concentrations of 100 nM at a flow rate of 5 μl/min to reach a low surface density corresponding to 130–190 RU (Table 4). The FKs were immobilized in flow cells 2, 3 and 4, which allowed measurements for all FKs in parallel. The empty flow cell 1 served as reference. For kinetic analysis in single cycle mode, 5 solutions of a 1+1 dilution series of integrins, starting at 20–40 nM, in HBS containing 1 mM of divalent metal ions ($CaCl_2$, $MgCl_2$, or $CaCl_2/MnCl_2$) were injected consecutively for 120 s at a flow rate of 30 μl/min. The dissociation was monitored for 300–600 s. For data evaluation, the reference channel and a series of buffer injections were subtracted (double referenced measurement). Sensorgrams were fitted with a 1:1 kinetic model using Biacore 2.0.3 evaluation software.

## Virus binding and FK saturation binding assays

For virus binding assays, $1x10^5$ trypsin-detached cells were washed and resuspended in modified FACS incubation/wash buffer and incubated for 1 h on ice using M1-/M3-IX-G and control M2-ΔE1A-G supernatant at a MOI 4 or control medium without virus. Cells detached with EDTA (see Flow cytometry) were alternatively used, but revealed no difference in virus binding. Divalent ions in the modified FACS buffer and in the incubation mix were adjusted to either $Mg^{2+}/Ca^{2+}$, 1 mM each, 1/0.2 mM $Mn^{2+}/Ca^{2+}$, or EDTA 2.5 mM. Subsequently, cells were incubated for 1 h on ice using saturating amounts of rabbit-anti-FK-M3 and secondary PE conjugate antibodies, followed by determination of bound virus by flow cytometry. For virus-mediated integrin antibody binding blocking assays, virus pre-incubated cells were subsequently incubated with αvβ6/αvβ8 antibodies and secondary PE conjugate antibodies.

For FK saturation cell-binding assays, cells prepared as above were incubated for 1 h on ice using a 4-fold dilution series of FKs ranging from 0.95 pM to 1 μM, or 3-fold series ranging from 0.47 pM to 83.33 nM. This was followed by addition of saturating amounts of rabbit-anti-FK-M3 and secondary PE conjugate antibodies, and determination of bound FK levels by flow cytometry. Parental B16 cells were included as a control in order to estimate and subtract fluorescence levels arising from low-affinity binding to other integrins and nonspecific background binding. Likewise, 2.5 mM EDTA-treated CMT-93 and M000216 cells were included to subtract nonspecific background binding. Apparent dissociation constants derived from the binding data were determined by Scatchard plot analysis as described in [133].

## Simulation of ternary complexes of mouse αvβ6 and αvβ8 with RGDLXXL peptides from FK-M1, FK-M3 and FMDV2

Querying the PDB database with the sequences from mouse integrin αv (Uniprot entry P43406), -β6 (Q9Z0T9) and -β8 (Q0VBD0) revealed several experimentally determined structures of binary and ternary integrin complexes. From this list of putative modeling templates we selected the PDB entries 5ffo (human integrin αvβ6/pro-TGFβ1 complex, 3.49 Å resolution, doi: 10.1038/nature21035), 6uja (human integrin αvβ8/pro-TGFβ1 complex, 3.3 Å resolution, doi: 10.1016/j.cell.2019.12.030) and 4wk0 (human integrin α5β1/RGD complex, 1.78 Å resolution, doi: 10.1073/pnas.1420645111), because of high sequence identity with the query sequences, resolution and length of the bound peptide. High sequence identity among query and template integrin sequences (S4 Table) suggests sufficient structural similarity for comparative modeling. Structural homology of the RGD peptides comes from the observation that pro-TGFβ1 residues (PDB id: 5ffo, residues D209-D228) fold into a βα-motif, which is similar

to the βα-motif seen in the foot-and-mouth disease virus structure (PDB id: 1fod, residues 139–158, doi: 10.1038/362566a0). Both βα-motifs are superimposable with a rmsd of 2.6 Å).

The mouse integrin complex structures were modeled using program Modeller version 10.1 [134]. Briefly, Modeller was used to generate a structure-based sequence alignment from the template structures. The mouse integrin sequences were subsequently aligned against this profile. This target-template alignment was manually curated for obvious alignment errors, particularly unnecessary gaps at the termini of the integrin chains were removed. The optimized alignments were used to generate 5 models of mouse integrin complexes and the model with the lowest value for the Modeller objective function was used for further evaluation.

## Supporting information

**S1 Dataset. Molecular docking simulation of A20M1 onto mouse integrin αvβ6.** The mouse integrin complex structures were modeled as described in Materials and Methods resulting in complex mαvβ6-A20M1.
(PDB)

**S2 Dataset. Molecular docking simulation of A20M3 onto mouse integrin αvβ6.** The mouse integrin complex structures were modeled as described in Materials and Methods resulting in complex mαvβ6-A20M3.
(PDB)

**S3 Dataset. Molecular docking simulation of A20FMDV2 onto mouse integrin αvβ6.** The mouse integrin complex structures were modeled as described in Materials and Methods resulting in complex mαvβ6-A20FMDV2.
(PDB)

**S4 Dataset. Molecular docking simulation of A20M1 onto mouse integrin αvβ8.** The mouse integrin complex structures were modeled as described in Materials and Methods resulting in complex mαvβ8-A20M1.
(PDB)

**S5 Dataset. Molecular docking simulation of A20M3 onto mouse integrin αvβ8.** The mouse integrin complex structures were modeled as described in Materials and Methods resulting in complex mαvβ8-A20M3.
(PDB)

**S6 Dataset. Molecular docking simulation of A20FMDV2 onto mouse integrin αvβ8.** The mouse integrin complex structures were modeled as described in Materials and Methods resulting in complex mαvβ8-A20FMDV2.
(PDB)

**S1 Fig. Genome maps of novel recombinant mouse adenoviruses.** The genomes of M1, M2 and M3 consist of five early transcription units, E1A, E1B, E2, E3, and E4 (red), two delayed early units IX and IVa2 (green), and one late unit, the major late unit giving rise to five major transcripts L1 to L5 (blue). The direction and lengths of transcription units are shown as solid arrows, relative to their position and orientation. In M1-IX-G and M3-IX-G, the 2A-GFP was inserted in frame at the end of the protein IX gene. In M1-ΔE1A-G, M2-ΔE1A-G and M3-ΔE1A-G, the E1A region was replaced with GFP-pA.
(EPS)

**S2 Fig. Genome maps of novel recombinant human adenoviruses.** The genomes of H5 and H35 consist of five early transcription units, E1A, E1B, E2, E3, and E4 (red), two delayed early

units IX and IVa2 (green), and one late unit, the major late unit giving rise to five major transcripts L1 to L5 (blue). The direction and lengths of transcription units are shown as solid arrows, relative to their position and orientation. H5-ΔE3B-CG, H5-ΔE3B-CG-FK-M1 and H5-ΔE3B-CG-FK-M3 contain a CMV-GFP-pA cassette replacing the deleted E3B region, in addition to a FK exchange from endogenous FK-H5 (black) with FK-M1 (red) and FK-M3 (blue), respectively. H35-ΔE1-CG contains a CMV-GFP-pA cassette replacing the deleted E1 region, plus an exchange of endogenous E4orf6 with H5-E4orf6.
(EPS)

**S3 Fig. Infectivity analysis of wt and recombinant M2 and M3 viruses.** (A) Immunoblot analyses of mouse CMT-93 cells infected with M2 wt and M2-ΔE1A-G using an MOI of 3. Cell lysate samples were harvested at six time points and analyzed with the indicated rabbit antibodies raised against early E1A-M2 and late M2-hexon protein, plus mouse antibodies against GFP and control actin. (B) Immunoblot analyses of mouse CMT-93 cells infected with M3 wt and M3-IX-G using an MOI of 3. Cell lysate samples were harvested at six time points and analyzed with the indicated rabbit antibodies raised against early E1A-M3 and late FK-M3, with the cross-reactive rabbit anti-hexon protein-M1, plus mouse antibodies against GFP and control actin. Staining with GFP-specific antibodies revealed two major processing forms, corresponding to processed GFP (Mr 27 kDa), and unprocessed IX-2A-GFP, (Mr 39.8 kDa), respectively.
(EPS)

**S4 Fig. Inhibition of MAdV and fiber-chimeric GFP reporter virus transduction in CMT-93 and M000216 cells by recombinant MAdV FKs.** (A-D) CMT-93 and (E) M000216 cells were preincubated for 1 h on ice using 5-fold dilution series of the indicated FK proteins starting with 800 ng/ml as highest concentration, followed by addition of the different GFP-expressing vectors and transfer to 37˚C for 48 h. FKb represents the biotinylated form of this protein. The viruses included H5-ΔE3B-CG (A), H5-ΔE3B-CG-FK-M3 (B), M1-IX-G (C), M3-IX-G (D), all at MOI 1, and M3-IX-G (E) at MOI 3. GFP analysis was performed 48 h pi, and expression index was normalized to FK-H3 control protein. For all experiments data represent biological triplicates, shown as mean ± SEM.
(EPS)

**S5 Fig. Quality control of knob proteins using SEC-MALS.** Proteins were separated on a Superdex200 size exclusion column and their molar mass was determined using RI and light scattering signals (see left y-axis). Chromatograms shown were determined at UV280 nm and plotted against the retention volume.
(EPS)

**S6 Fig. Inhibition of M2-ΔE1A-G reporter virus transduction in CMT-93 cells by anti-FK antisera.** The M2-ΔE1A-G virus was pre-incubated for 1 h at RT with serial 5-fold dilutions of the rabbit anti-FK-M2 and control anti-FK-M3 and -FK-H3 antisera ranging from 1:1,250 to 1:781,250, followed by addition of the mixes to CMT-93 cells for 48 h at 37˚C. The virus input amounted to an MOI 1, and samples were processed for analysis as described for Fig 2F and 2G.
(EPS)

**S7 Fig. Evaluation of human CD46 and DSG-2 as species cross-reactive receptor for M1/M2/M3.** (A) Red and blue histograms show cytofluorometric analysis of human CD46 and DSG-2 in CHO, CHO-hCD46, CHO-hDSG-2, and control human A549 cells, respectively. The grey histograms show background staining obtained using an isotype control. Numbers

indicate MFI values of specific or control antibodies. (B) Parental CHO, CHO-hCD46, CHO-hDSG-2 and A549 cells were infected with M1-IX-G, M2-ΔE1A-G, M3-IX-G and H3-ΔE1-CG and H35-ΔE1-CG using an MOI 50 for all CHO cell types, and an MOI 10 for A549 cells. Parental CHO cells have low and human A549 cells have high sensitivity for members of the HAdV- B species. Cells were analyzed by flow cytometry 48 h pi. MFI values for GFP in red and blue histograms are from uninfected and infected cells, respectively.
(EPS)

**S8 Fig. Inhibition of virus infection by β6- and β8-specific antibodies.** Virus infection interference in CMT-93 (A), B16-mβ6 (B), B16-mβ8 (C) and M000216 cells (D) by β6- and β8-specific antibodies. Cells were pre-incubated for 1 h on ice using 5-fold dilution series of the indicated integrin antibodies starting with 800 ng/ml as highest concentration, followed by addition of the different indicated GFP-expressing viruses and transfer to 37˚C for 48 h. The virus input amounted to an MOI 1 for CMT-93 cells, and MOI 3 for the B16 type and M000216 cells. Cells were further cultivated and processed as described for Fig 6B and 6C. Data represent biological triplicates, shown as mean ± SEM.
(EPS)

**S9 Fig. Inhibition of virus infection by sITGs.** Infection blocking assays by sITGs. M3-IX-G virus was incubated for 1 h at RT with a 5-fold serial dilution of the indicated sITGs starting from 800 to 6.4 ng/ml or to 1.28 ng/ml, followed by addition to CMT-93 cells (A) and M00216 cells (B). Alternative sITG hαvβ82 was obtained from a different company. Cells were further cultivated and processed as described for Fig 6D and 6E. Data represent biological triplicates, shown as mean ± SEM.
(EPS)

**S10 Fig. Quality control sITGs by PAGE analysis.** Two μg of the indicated eight sITGs were size-fractionated under reducing conditions on a 10% PAGE, followed by PAGE-blue staining.
(EPS)

**S11 Fig. Inhibition of virus infection by peptides.** Cells were pre-incubated on ice with either 5-fold serial dilutions of peptides resulting in final concentrations from 5,000 to 0.32 nM, or at the highest concentration of 5,000 nM only. Viruses were added to cells, followed processing as described for Fig 6G and 6H. Virus/cell combinations included: M3-IX-G/CMT-93 (A), H5-ΔE3B-CG-FK-M1/CMT-93 (B), M2-ΔE1A-G/CMT-93 (C), H5-ΔE3B-CG/CMT-93 (D), and M3-IX-G/M000216 (E). Data represent biological triplicates, shown as mean ± SEM.
(EPS)

**S12 Fig. FK affinity determination to human sITG αvβ6 and αvβ8 by SPR analysis.** Sensor chips containing immobilized FK-M1 and FK-M3 were probed with human sITG αvβ6 (A, B), or human sITG αvβ8 (C, D). Following consecutive analyte injections over 120 s, dissociation was monitored for 600 s (black). Sensorgrams were fitted with a 1:1 kinetic model (red).
(EPS)

**S13 Fig. FK affinity determination to cell surface integrins by saturation binding.** FK saturation cell binding assays using cells with defined αvβ6/αvβ8 expression included FK-M3 binding to CMT-93 (A), and FK-M3 binding to M000216 (B) respectively. Cells treatment with 2.5 mM EDTA was included in order to subtract background levels when calculating equilibrium dissociation constant $K_D$ values by Scatchard plot analyses.
(EPS)

**S14 Fig. Examples of Scatchard plot analyses of FK-M1/-M3 high-affinity binding to αvβ6/αvβ8-expressing cells.** Background-subtracted binding isotherms and Scatchard plot analyses, respectively, are shown for FK-M1 interaction to B16-mβ6 cells (A, B, corresponding to data in Fig 7C), for FK-M3 interaction to CMT-93 cells (C, D, corresponding to data in S13A Fig), for FK-M3 interaction to B16-mβ6 cells (E, F, corresponding to data in Fig 7D), for FK-M3 interaction to B16-mβ8 cells (G, H, corresponding to data in Fig 7E), and for FK-M3 interaction to M000216 cells (I, J, corresponding to data in S13B Fig). Background-subtracted FK binding values of the first binding phase used for Scatchard plot analyses are denoted by black symbols.
(EPS)

**S1 Table. Determination molar mass and mass fraction of knob preparations by SEC-MALS.**
(DOCX)

**S2 Table. Oligonucleotides used for generation of BACmid and knock out constructs.**
(DOCX)

**S3 Table. Recombinant protein data and oligonucleotides used for generation of expression constructs.**
(DOCX)

**S4 Table. Target-template sequence identity.**
(DOCX)

**S1 Text. Cell library screening.**
(DOCX)

**S2 Text. Generation recombinant viruses.**
(DOCX)

## Acknowledgments

We thank all providers of wt viruses, cell lines, antibodies and plasmid reagents, Leta Fuchs for technical assistance, the Flow Cytometry Facility of the University of Zurich Irchel for their help with flow cytometry and cell sorting. We are grateful to Maarit Suomalainen, Silke Stertz and Stefan Kochanek for the useful discussion of the data, Sergio Gloor for help with Scatchard plot analyses, and Izadora Demmer, Sandra Ramelli, Balthasar Müller, and Marco Amsler for their contributions to the diverse virus constructs, and Katherine Spindler for helpful comments on the manuscript.

## Author Contributions

**Conceptualization:** Manuela Bieri, Rodinde Hendrickx, Urs F. Greber, Silvio Hemmi.

**Funding acquisition:** Manuela Bieri, Urs F. Greber, Silvio Hemmi.

**Investigation:** Manuela Bieri, Rodinde Hendrickx, Michael Bauer, Bin Yu, Tania Jetzer, Birgit Dreier, Peer R. E. Mittl, Jens Sobek, Silvio Hemmi.

**Methodology:** Manuela Bieri, Rodinde Hendrickx, Michael Bauer, Bin Yu, Birgit Dreier, Peer R. E. Mittl, Jens Sobek, Silvio Hemmi.

**Project administration:** Silvio Hemmi.

**Resources:** Manuela Bieri, Rodinde Hendrickx, Michael Bauer, Bin Yu, Birgit Dreier, Peer R. E. Mittl, Jens Sobek, Andreas Plückthun, Urs F. Greber, Silvio Hemmi.

**Supervision:** Urs F. Greber, Silvio Hemmi.

**Validation:** Manuela Bieri, Rodinde Hendrickx, Silvio Hemmi.

**Writing – original draft:** Manuela Bieri, Rodinde Hendrickx, Silvio Hemmi.

**Writing – review & editing:** Manuela Bieri, Rodinde Hendrickx, Michael Bauer, Bin Yu, Tania Jetzer, Birgit Dreier, Peer R. E. Mittl, Jens Sobek, Andreas Plückthun, Urs F. Greber, Silvio Hemmi.

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
