## [Decision Letter · Decision Letter 0]

27 Sep 2021

Dear Dr. Hemmi,

Thank you very much for submitting your manuscript "The RGD-binding Integrins avb6 and avb8 are Receptors for Mouse Adenovirus-1 and -3 Infection" for consideration at PLOS Pathogens. As with all papers reviewed by the journal, your manuscript was reviewed by members of the editorial board and by several independent reviewers. The reviewers appreciated the attention to an important topic. Based on the reviews, we are likely to accept this manuscript for publication, providing that you modify the manuscript according to the review recommendations.  Please be sure to address each of the points that ate outlined in the reviews provided.  In addition, Reviewer 2 suggests one additional experiment that would strengthen the current manuscript (see comments provided below) that should be considered and addressed.  

Sincerely,

Donna Neumann, PhD

Associate Editor

PLOS Pathogens

Karl Münger

Section Editor

PLOS Pathogens

Kasturi Haldar

Editor-in-Chief

PLOS Pathogens

orcid.org/0000-0001-5065-158X

Michael Malim

Editor-in-Chief

PLOS Pathogens

orcid.org/0000-0002-7699-2064

Reviewer Comments (if any, and for reference):

Reviewer's Responses to Questions

**Part I - Summary**

Reviewer #1: Hemmi et al. have identified and characterized the receptors mediating infection of the mouse adenovirus (MAdV) using novel recombinant mouse viruses. They determine that RGD-recognizing ⍺vß6 and ⍺vß8 integrin heterodimers are the receptors for MAdV types M1 and M3. This is based on six different experimental criteria including studies in permissive/nonpermissive cell lines, gain-of-function in non-permissive cells, antibody-mediated blocking, synthetic peptide-mediated blocking, CRISPR/shRNA knockout/knockdown studies and SPR analysis. They provide substantial associative, deductive and experimental evidence to support their findings. The studies are rigorously performed, thoroughly reported and are of suitable novelty to merit publication in PLOS Pathogens (provided they address the minor comments below).

Reviewer #2: This study identifies a cell entry receptor for mouse adenoviruses MAdV M1 and M3. Little is known about entry of murine adenoviruses, and identification of receptors would expand our understanding of infection and could have implications for gene therapy vector delivery. A previous report from the Spindler group suggested that alpha-v integrins and heparan sulfate are used. The authors here focus on integrins as potential receptors for M1 and M3 and they start by generating new reporter viruses. They used these viruses to screen cells for infection. Recombinant fiber knob and antisera blocked transduction by these reporter viruses and suggested similar receptor usage for M1 and M3. They show that the CAR receptor is not used and then demonstrate that overexpression of integrins enhanced binding and infection. Anti-integrin antibodies blocked M1 and M3 binding and infection. Infection was also blocked by soluble integrin, and RGDLXXL peptides. They also showed that genetic depletion of integrins reduced infection. Direct binding of fiber to integrins was demonstrated by surface plasmon resonance and cell binding. Finally, molecular docking simulations support that the RGD sequence in the fiber protein interacts with the mouse integrin chain. They conclude that the RGDL motif in the fiber knob is important for M1 and M3 infection through the conserved alpha-v-beta-6/8 integrins. The work is comprehensive, rigorous and compelling. The Introduction and Discussion are well written, but the Results section is a little challenging to work through. Overall, this is a very nice contribution to the field, and I have little to suggest for improvement.

**Part II – Major Issues: Key Experiments Required for Acceptance**

Reviewer #1: I have no major comments or concerns about this manuscript.

Reviewer #2: Having identified the RGD motif in the fiber as important for interaction with integrins, the work could be further strengthened by showing that mutant peptides cannot bind to the integrin in the SPR and cell binding studies of Figure 7, and also cannot inhibit transduction. This would close the loop and further support their conclusions.

**Part III – Minor Issues: Editorial and Data Presentation Modifications**

Reviewer #1: Figure 1A: The actin levels in the immunoblots suggest that the samples are not loaded consistently. At 72 hpi timepoints this might be due to cytopathic effects, but can the authors provide an explanation for the inconsistency in loading at 8 hpi and 14 hpi timepoints for M1-IX-G?

Figure 1B, 1C: Although the authors note that the data presented is in triplicate and are presented as mean +/- SEM, there are no error bars or statistics visible on these graphs.

Line 115-117: This seems like a contradictory statement. Please restate or clarify

Figure 2: While the trendlines and error bars are somewhat compelling, the authors should provide information on statistical significance of this datasets to remove any ambiguity around the conclusions.

Figure 5B/C: Please consider spacing out the labels containing the ion names (Mg/CaMn/Ca..etc) to increase clarity.

Where is figure 6J? The figure and labels jump from 6I to 6K.

Line 504: “undermined”, as stated in the manuscript means “to erode”. I think the authors mean “underlined” (meaning, “to highlight”).

Line 607: misspelled “criteria”

Reviewer #2: 1) The recombinant viruses generated are useful tools for the field. It would be helpful to include maps for the recombinant viruses generated in Table 1.

2) The Introduction is well written, but I suggest one addition. The definition of a viral receptor is sometimes controversial. I suggest the authors add a section in which they provide their definition for a viral receptor and the criteria that need to be met. They then can highlight how they go onto to fulfil all these requirements.

PLOS authors have the option to publish the peer review history of their article (what does this mean?). If published, this will include your full peer review and any attached files.

Reviewer #1: No

Reviewer #2: **Yes: **Matthew D. Weitzman

Figure Files:

Data Requirements:

Reproducibility:

References:

---

## [Editor Report · Decision Letter 1]

1 Nov 2021

Dear Dr. Hemmi,

We are pleased to inform you that your manuscript 'The RGD-binding Integrins avb6 and avb8 are Receptors for Mouse Adenovirus-1 and -3 Infection' has been provisionally accepted for publication in PLOS Pathogens.

Best regards,

Donna Neumann, PhD

Associate Editor

PLOS Pathogens

Karl Münger

Section Editor

PLOS Pathogens

Kasturi Haldar

Editor-in-Chief

PLOS Pathogens

orcid.org/0000-0001-5065-158X

Michael Malim

Editor-in-Chief

PLOS Pathogens

orcid.org/0000-0002-7699-2064
---

## [Editor Report · Acceptance letter]

26 Nov 2021

Dear Dr. Hemmi,

We are delighted to inform you that your manuscript, "The RGD-binding Integrins avb6 and avb8 are Receptors for Mouse Adenovirus-1 and -3 Infection," has been formally accepted for publication in PLOS Pathogens.

Best regards,

Kasturi Haldar

Editor-in-Chief

PLOS Pathogens

orcid.org/0000-0001-5065-158X

Michael Malim

Editor-in-Chief

PLOS Pathogens

orcid.org/0000-0002-7699-2064